# Epidemiological and ecological consequences of virus manipulation of host and vector in plant virus transmission

**Nik J. Cunniffe**[1]*, **Nick P. Taylor**[1], **Frédéric M. Hamelin**[2], **Michael J. Jeger**[3]

**1** Department of Plant Sciences, University of Cambridge, Cambridge, United Kingdom, **2** IGEPP, INRAE, Institut Agro, Univ Rennes, Rennes, France, **3** Department of Life Sciences, Imperial College London, Ascot, United Kingdom

* njc1001@cam.ac.uk

## Abstract

Many plant viruses are transmitted by insect vectors. Transmission can be described as persistent or non-persistent depending on rates of acquisition, retention, and inoculation of virus. Much experimental evidence has accumulated indicating vectors can prefer to settle and/or feed on infected versus noninfected host plants. For persistent transmission, vector preference can also be conditional, depending on the vector's own infection status. Since viruses can alter host plant quality as a resource for feeding, infection potentially also affects vector population dynamics. Here we use mathematical modelling to develop a theoretical framework addressing the effects of vector preferences for landing, settling and feeding–as well as potential effects of infection on vector population density–on plant virus epidemics. We explore the consequences of preferences that depend on the host (infected or healthy) and vector (viruliferous or nonviruliferous) phenotypes, and how this is affected by the form of transmission, persistent or non-persistent. We show how different components of vector preference have characteristic effects on both the basic reproduction number and the final incidence of disease. We also show how vector preference can induce bistability, in which the virus is able to persist even when it cannot invade from very low densities. Feedbacks between plant infection status, vector population dynamics and virus transmission potentially lead to very complex dynamics, including sustained oscillations. Our work is supported by an interactive interface https://plantdiseasevectorpreference.herokuapp.com/. Our model reiterates the importance of coupling virus infection to vector behaviour, life history and population dynamics to fully understand plant virus epidemics.

**Data Availability Statement:** The source code for our interactive online interface is available at https://github.com/nt409/vector-interface; this

## Author summary

Plant virus diseases–which cause devastating epidemics in plant populations worldwide–are most often transmitted by insect vectors. Recent experimental evidence indicates how vectors do not choose between plants at random, but instead can be affected by whether plants are infected (or not). Virus infection can cause plants to "smell" different, because they produce different combinations of volatile chemicals, or "taste" different, due to

Python code also acts as a reference implementation of our mathematical model.

**Funding:** NPT acknowledges general support from the Biotechnology and Biological Sciences Research Council of the United Kingdom (BBSRC https://bbsrc.ukri.org/) via a University of Cambridge DTP PhD studentship (Project Reference 2119688). The funders had no role in study design, data collection and analysis, decision to publish, or preparation of the manuscript.

**Competing interests:** The authors have declared that no competing interests exist.

chemical changes in infected tissues. Vector reproduction rates can also be affected when colonising infected versus uninfected plants. Potential effects on epidemic spread through a population of plants are not yet entirely understood. There are also interactions with the mode of virus transmission. Some viruses can be transmitted after only a brief probe by a vector, whereas others are only picked up after an extended feed on an infected plant. Furthermore there are differences in how long vectors remain able to transmit the virus. This ranges from a matter of minutes, right up to the entire lifetime of the insect, depending on the plant-virus-vector combination under consideration. Here we use mathematical modelling to synthesise all this complexity into a coherent theoretical framework. We illustrate our model via an online interface https://plantdiseasevectorpreference. herokuapp.com/.

## Introduction

Plant diseases have impacts on crops, affecting yield, and on natural plant ecosystems and landscapes, affecting population and community structure [1–3]. Plant diseases caused by viruses can be particularly damaging [4]. Most plant viruses are transmitted by vectors, which both acquire virus from infected plants and inoculate virus to healthy plants. These vectors can come from a range of insect families, although aphids, whiteflies and other hemipterans with piercing mouthparts are particularly important [5]. Transmission plays a critical part in disease epidemiology and depends to a large extent on vector life history and behaviour concerning movement, landing, settling, feeding, and reproduction on plants. These aspects are directly linked to the mode of transmission, whether non-persistent, semi-persistent, or persistent [6]. In non-persistent transmission the virus is restricted to the insect's stylet, in semi-persistent transmission the virus enters the insect's foregut, and in persistent transmission the virus passes through the gut to the haemolymph and then to the salivary glands. These modes of transmission can be characterised in large part by the rates of vector acquisition and inoculation and the retention time within the vector [7]. The distinction between non-persistent (transient) and persistent (intimate) transmission is particularly marked. Hence, attempts to determine what drives a plant virus epidemic should not only aim to characterise the plant-virus interaction, but also the plant-vector and virus-vector interactions.

Epidemiological models have often assumed that vector transmission occurs essentially at random [8]. However, given the ongoing accumulation of experimental evidence suggesting more complexity, such an assumption when used in developing mathematical models is often an over-simplification [4]. Vectors show preferences in their interactions with plants that can directly influence the rate of transmission. Preference can take many forms, with much past work focusing on arthropod vectors of vertebrate-host parasites [9–12]. Focusing on plant viruses in particular, vector preference can be expressed in different ways: (i) host preference within a vector's host range; (ii) preference for the host phenotype, restricted in the sense here to whether the host is infected or healthy; and (iii) conditional preference, whereby the preference for infected or healthy hosts depends on whether the vector is viruliferous or non-viruliferous.

Plant viruses are often termed as generalist or specialist in relation to their host range. Yet, such classification makes little sense for insect-transmitted plant viruses unless account is taken of the vector life history, host preferences (for landing, settling, feeding and reproduction) and transmission characteristics. Host range preferences have been well characterised for many insect vectors, but there is not always a simple relationship with insect life history [13] or

plant traits [14]. The second aspect of vector preference we identify, host phenotype preference, refers to the vector's preference for infected or healthy hosts, irrespective of whether the vector is non-viruliferous or viruliferous. Preferences for landing and feeding are dependent on sensory cues which can be olfactory as well as visual [15]. However, as pointed out by Sisterson [16], it is important to distinguish landing, or orientation, preference from feeding preference, since both can potentially affect pathogen spread differently.

For non-persistent transmission, it seems likely that inhibition of settling while allowing probing would encourage transmission, whereas prolonged settling would retard transmission. However, mathematical modelling indicates that both inhibition and prolonged settling can contribute to epidemic development, for example via density dependent production of alate (winged) forms on crowded host plants [17,18]. For persistent transmission, thrips species have long been reported to have settling and feeding preferences for tospovirus-infected plants. Such preferences can affect thrips life history traits. Virus acquisition occurs with early instars and the developmental period to adults was shortened when *Thrips palmi* had acquired virus from groundnut bud necrosis virus-infected plants [19]. Such a shortening may facilitate the early inoculation of healthy plants by inoculative adults. The effects of volatile emissions on vector behaviour cannot always be extrapolated from one species to another [20]. More *Bemisia tabaci* whiteflies landed and settled on plants infected with squash vein yellowing virus than non-infected squash, but the opposite was found with watermelon [21]. Whitefly behaviour differed between the two cucurbit hosts but integrating the various life history traits into a comparison of potential disease dynamics suggested a more rapid spread of the virus in watermelon fields.

Where a virus modifies plant quality sufficiently to improve it as a feeding resource, this can lead to an accompanying increase in vector abundance termed as pathogen-mediated insect superabundance [22]. However, vector preferences are not always consistent across different hosts, even when virus-infected plants are generally higher quality hosts [23,24]. The effect of insects feeding on plants infected with viruses they do not vector has also been studied. When *B. tabaci* preferentially feeds on different hosts infected with tomato spotted wilt virus (which is thrips-vectored), body size, longevity and fecundity were all reduced [25] indicating that the initial preference for a virus-infected plant was induced by host volatiles and not by subsequent performance. However, host volatiles are not always the cue for settling and feeding preference. For the beetle-transmitted bean pod mottle virus, the beetles are more attracted to infected soybean plants, which have dramatically higher sucrose levels, and although beetles consumed less leaf foliage per plant, they fed on more plants per unit of time when they had high levels of sucrose [26]. Pea aphid clones adapted to either pea or alfalfa were tested to see how bean leafroll virus affected their performance and preference [27]. Aphid clone x host plant species x virus status interactions and unique virus-association phenotypes were found.

Effects on vector fitness can also be seen, for example, on squash infected with papaya ringspot virus, transmitted by *Aphis gossypii* [28]. The overall performance of *A. gossypii* was substantially higher, with an extended settling (arrestment), on infected plants when compared to on healthy plants. This effect was not present with the non-vector *B. tabaci*. Hence there was a positive fitness effect on the vector in what would be considered a transient vector-virus interaction. Equally, effects on plant fitness can be negative in the longer term [29]. It was found that initial feeding of *Myzus persicae* on *Nicotiana tabacum* infected with cucumber mosaic virus reduced the reproduction rate and longevity of aphids subsequently introduced to the previously foraged plants.

As well as affecting feeding, virus-induced changes in host phenotype can also affect vector dispersal and disease spread, depending on transmission mode and efficiency [30]. In studies

of turnip yellows virus on *Montia perfoliata* transmitted by *Brevicoryne brassicae* in the persistent mode [31], it was found that non-viruliferous aphids showed greater fecundity only on infected plants and had reduced dispersal and activity levels. However, viruliferous aphids showed greater dispersal and activity levels with a greater fecundity and efficiency in feeding irrespective of plant infection status.

Just as plant viruses can modify the host phenotype and affect insect vector behaviour and population dynamics, modification of the vector phenotype so that preference is conditional on whether the vector is non-viruliferous or viruliferous can also affect disease dynamics. For conditional vector preference, the question arises: whether, and under what circumstances, the virus is manipulating the plant host and the vector to its own advantage? If this is indeed the case and it can be shown that there is a genetic basis to such manipulation, then the question takes on an evolutionary as well as epidemiological dimension. The "Virus manipulation hypothesis" that changes in vector behaviour are induced by both host and vector phenotype was proposed by Ingwell et al. [32]. Accordingly, non-persistent and persistent transmission of viruses will have different effects on vector preferences for landing, settling, feeding, and dispersal from infected and healthy plants [33–35]. Any difference in reported effect seems to depend on whether probing or settling was evaluated in studies [36,37]. Overall, the evidence supports the view that viruses manipulate both host and vector to enhance transmission [38] and hence fitness. A substantial body of evidence was reviewed by Eigenbrode et al. [39], albeit with some inconsistency across findings (see Tables 1 and 2 and Fig 1 both of [39]). This inconsistency may be due to a lack of occurrence or, despite the large number of studies reviewed, a lack of information. There appears to be a remarkable degree of convergence among unrelated viruses with similar transmission characteristics [38,40]. Predictions were tested for non-persistent, semi-persistent and persistent transmission, with adaptive manipulation most apparent in the latter case. However, although the main factors influencing transmission and selection for manipulative traits have been identified, there are important gaps in linking findings with evolutionary processes [41], especially the molecular and environmental constraints on virus manipulation. A further constraint is the difficulties of exploring the consequences of virus manipulation, where this has been shown in laboratory or microcosm studies, in field settings in crops or wild plant populations. Of course, mathematical modelling presents one way of dealing with this constraint [42,43].

Mathematical modelling has contributed to analysis of the epidemiological consequences of vector preferences [16]. Previously, in modelling the spread of barley yellow dwarf virus, it was found that with a low incidence of infected plants, disease spread is favoured by vectors preferring infected plants, whereas with a high incidence, spread is favoured by vectors favouring healthy plants [44]. However, this initial model did not distinguish the preferences of viruliferous and non-viruliferous vectors. Preferences of vectors for healthy or infected host plants will affect both aggregation and dispersal of vectors [45]. Assumptions on birth and death rates of vectors based on their infectivity status can have important effects on both vector population dynamics and should be included in epidemiological models [46]. Including conditional preference depending on vector phenotype into models, showed that a switch in preference once a vector acquires virus from infected plants can enhance spread [47].

A more comprehensive model with conditional vector preferences, but also including more vector life history traits, especially dispersal, has also been developed ([48] as corrected in [49]). Traits including intrinsic growth rate, population carrying capacity, and landing and departure rates, were introduced in Shaw et al. [48] conditional upon whether the host is healthy or infected and whether the vector is viruliferous or non-viruliferous. The form in which preference was introduced in this model, different to Roosien et al. [47], used a phenomenological response in which the fraction of infected plants was raised to a power

characteristic of the vector preference. The Shaw et al. [48] model of vector dynamics distinguishes the density of vectors currently colonising infected vs. uninfected plants, allowing intraspecific competition between vectors at the scale of the individual host to be tracked, as well as how that depends on plant infection status. The model also allows density-dependent effects on vector dispersal rates to be distinguished, as well as–again–how this depends on plant infection status. The model was parameterised for barley yellow dwarf virus and potato virus Y, although–given the interaction between density and rate in the transmission terms–it is difficult to follow from what is presented in the paper how this was done, precisely. The key result from numerous simulations indicated that vector population growth rates overall had the greatest effect on virus spread, with rates of vector dispersal from hosts of the same virus status as the vector also important. These interpretations were based mainly on numerical simulations of the model with a global sensitivity analysis based on a partial rank correlation coefficient technique for the time taken to reach 80% of hosts infected.

A further model describing vector preferential behaviour and how this affects transmission was developed specifically for the tospovirus tomato spotted wilt virus vectored by the thrips species *Frankliniella occidentalis* [50]. In this system there is persistent-propagative transmission which is also transstadial: acquisition only occurs at the larval stages, the virus replicates within the vector during the development stages, and mobile adults are then able to inoculate virus, as previously modelled [51]. Ogada et al. [50] included a linear relative preference term to reflect that viruliferous adults prefer healthy plants while non-viruliferous adults prefer infected plants. Thus, eggs which are laid on infected plants would likely result in larvae which acquire virus. In general, in this paper, we ignore the direct effects of the virus on vector birth and death rates, which are most apparent with viruses that propagate in the vector [52,53].

The objective in this paper is to develop a model specific to viral pathogens of plants vectored by insects, building on previous epidemiological models. The model is complex but includes biologically relevant parameters that can be estimated from experimental data, although these data are mostly obtained in microcosm studies rather than from field observations. The distinction between our model and the earlier models is that, following appropriate parameterisation, the model can be used for both non-persistently transmitted and persistently transmitted viruses. A further distinction is that we use analytical methods to derive epidemiological quantities that depend on the model parameters, and to show how model behaviours depend on parameter values. We derive the basic reproduction number and endemic equilibria analytically and use these results to show how transmission type and vector life-history and behaviour interact with conditional vector preference, as well as to determine under which conditions different long-term outcomes are possible depending on initial conditions. To illustrate the commonalities and differences in approaches with previous models of conditional vector preference, the variables and parameters defined are compared in S1 Appendix. We also introduce a user-friendly interactive online interface to our model, which offers readers of our paper an opportunity to understand for themselves how changes to underlying epidemiological parameters, as well as those relating to aspects of vector preference, lead to different epidemiological outcomes.

## Methods

### Overview of epidemiological approach

A model of plant-virus-vector interactions has been previously introduced to account for the range of transmission classes in plant-virus-vector interactions [54,55]. Here, we modify that model to account for the effects of conditional vector preference on life history from landing, settling, and feeding, to reproduction and flight. In addition, more detail on vector population

dynamics is included as an important element in virus epidemiology [56]. The following assumptions were made:

1. Vectors are attracted by plant cues (visual or olfactory) to land on infected plants to a greater or lesser extent.

2. Whether or not vectors then settle and feed for an extended period, or only probe and then depart, also depends on the plant's infection status.

3. The strength of vector preference can differ for viruliferous and non-viruliferous vectors, i.e. preference is conditional on vector status as well as plant infection status. We note that in our analysis of the model we assume that conditional preference is relevant only for persistent transmission; for non-persistent transmission we assume that viruliferous and non-viruliferous vectors behave similarly.

4. The proportion of probes that leads to vectors settling for an extended feed affects the number of plants visited by vectors per unit of time, and so the overall transmission rate.

5. Whether vectors probe or feed has different effects on transmission for non-persistent vs persistent viruses.

6. The fecundity of the vector can be affected by the plants it feeds on, with vectors that predominantly feed on infected plants potentially having either a higher or lower birth rate.

7. The loss rate of the vector, whether from additional mortality or the chance of movement away from the plant population, may be affected by the number of distinct plants visited per extended feed, reflecting the possibility that vectors must travel between plants more often if the number of plants visited per feed is increased.

8. The flight pattern of a vector (duration of flights) may depend on whether it is viruliferous or non-viruliferous.

In our model we account for the effects of different vector preferences for healthy susceptible vs. infected plants, potentially distinguishing different preferences of non-viruliferous vs. viruliferous vectors, as well as preference for probing plants vs. settling for an extended feed. As we describe below, much of the complexity of our model comes from properly accounting for the accumulative effects of these behaviours on: i) numbers of contacts between vectors and hosts per unit time, and ii) vector population dynamics.

## The basic model

We assume that there is no latent period in the plant or vector population, no propagative or transovarial transmission of the virus, and no immigration or emigration of vectors. The underlying model is (see also Fig 1):

$$\frac{dS}{dt} = \rho N - \Lambda(S, I, Z) - \rho S,$$

$$\frac{dI}{dt} = \Lambda(S, I, Z) - (\rho + \mu)I,$$

$$\frac{dX}{dt} = g(S, I, X, Z) + \tau Z - \Omega(S, I, X) - h_-(S, I)X,$$

$$\frac{dZ}{dt} = \Omega(S, I, X) - \tau Z - h_+(S, I)Z.$$

(1)

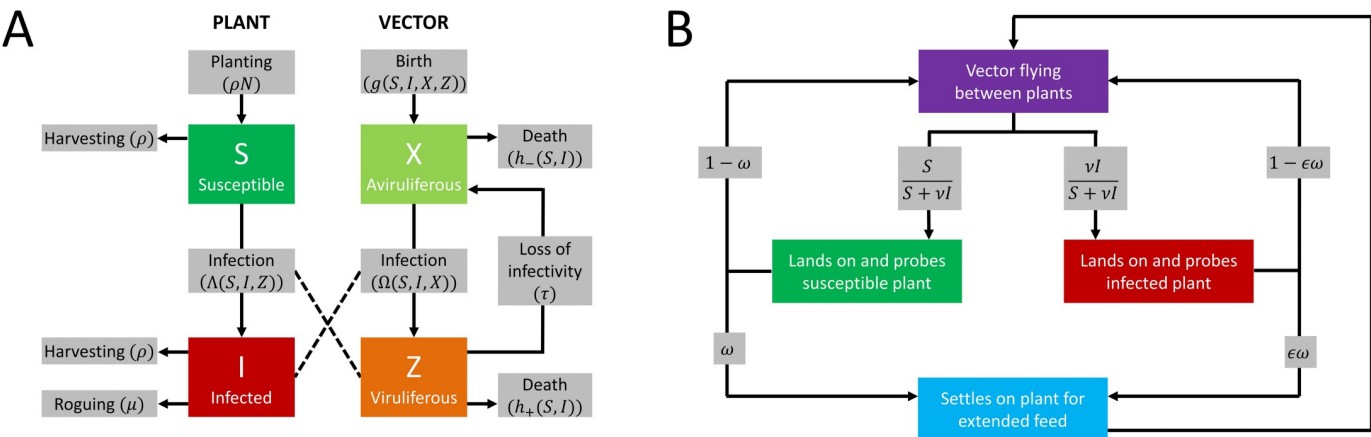

**Fig 1. Model structure. (A)** Compartments in the epidemiological model. All arrows corresponding to transitions causing hosts or vectors to leave a compartment are labelled with per capita rates, apart from those corresponding to infection of hosts or acquisition of the virus by vectors (i.e. from S→I and from X→Z) which are labelled with net rates. The rate of planting of hosts and birth of vectors are also labelled with net rates. **(B)** Schematic showing how the movement, settling, and feeding behaviours of vectors are modelled. These affect the infection terms (i.e. Λ and Ω) in the epidemiological model by controlling the probability of transmission (Eq (26)), as well as the average rate at which vectors visit plants (Eqs (11) and (13)). For persistent transmission (PT), the values of $v$, $\omega$ and $\epsilon$ can be different for viruliferous and non-viruliferous vectors (e.g. for a PT virus, the bias to land on infected plants is $v_-$ for non-viruliferous vectors, $v_+$ for viruliferous vectors). For non-persistent transmission (NPT), the values of these parameters do not depend on the vector's infection status. Note that in the schematic "flying" refers to any movement between plants: for vectors such as aphids that also–or exclusively–produce non-winged forms, this corresponds to movement between plants by crawling.

The variables $S$ and $I$ in Eq (1) are the densities (per unit area) of healthy susceptible and infected plants, respectively, with $N$ the plant population density in the absence of disease. The variables $X$ and $Z$ are the densities of non-viruliferous and viruliferous vectors respectively, with $\kappa$ the vector population density in the absence of disease. A list of all model variables and parameters in the model is given in Table 1. Plants are removed by harvesting or natural mortality at rate $\rho$ irrespective of disease status. Infected plants are also subject to an additional removal at rate $\mu$, which could correspond to disease-induced mortality or alternatively to roguing of infected plants (i.e. systematic removal of symptomatic plants). While more complex host growth functions would be possible and would potentially affect model dynamics (see e.g. [57]), for tractability here we take a relatively simple model of host planting in which losses in the plant population are replenished by replanting with susceptible plants at a fixed rate $\rho N$, i.e proportional to the population density in the absence of disease. The flexibility of the parameters $\rho$ and $\mu$ mean that the model could refer to a managed crop population or a natural plant population, and we note that since the rate of host birth is in general different to the net rate of natural/disease-induced host death that the current level of infection affects the host population density in the model. The rate at which viruliferous vectors cease to be inoculative is $\tau$. The model includes birth of vectors at rate $g(S,I,X,Z)$ with all vectors born non-viruliferous and in which the vector birth rate is averaged over the vector population and so depends on the fraction of plants that are healthy $S$ or infected $I$. Death of non-viruliferous and viruliferous vectors occurs at per capita rates $h_-(S,I)$ and $h_+(S,I)$ respectively, where these rates again account for the infection statuses of the plant population. The $S$ and $I$ dependences in vector birth and death rates allow the consequent effects of vector preference on vector population dynamics to be tracked.

Apart from the effects of vector preference on the vector's population dynamics, changes relative to the model of Jeger et al. [54] largely focus on altering the transmission rates to account for the preferences of non-viruliferous vs viruliferous vectors to land, probe and/or

**Table 1. Summary of symbols and their meanings, with default values used in numerical work.** NPT indicates non-persistent transmission, PT persistent transmission.

| Symbol | Meaning | Value |
|---|---|---|
| $S$ | Density of susceptible and healthy plants | - |
| $I$ | Density of infected plants | - |
| $X$ | Density of non-viruliferous vectors | - |
| $Z$ | Density of viruliferous vectors | - |
| $N$ | Density of plants in the absence of disease | 1,000 |
| $\rho$ | Rate at which plants die naturally/are harvested | 0.01 d$^{-1}$ |
| $\mu$ | Rate of roguing (or additional disease-induced death) | 0.01 d$^{-1}$ |
| $\alpha$ | Rate at which vectors die in the absence of any effect due to number of flights per feed | 0.12 d$^{-1}$ |
| $\tau$ | Rate at which vectors lose infectivity | 4 d$^{-1}$ (NPT) 0.05 d$^{-1}$ (PT) |
| $\sigma$ | Per capita rate at which vectors are born as the vector population density tends to zero | 0.18 d$^{-1}$ |
| $\zeta$ | Vector population density at which density dependence reduces the birth rate to zero | 1977.6 (NPT) 163.2 (PT) |
| $\Gamma$ | Average time spent feeding on a plant when vector chooses to settle for extended feed | 2 d |
| $\delta$ | Controls any increase in the death rate as more plants are visited per feed | Varied (Default = 0) |
| $\beta$ | Proportionate change in birth rate on infected plants | Varied (Default = 1) |
| $v_-$ | Bias of non-viruliferous vector to land on infected plants | Varied (Default = 1) |
| $v_+$ | Bias of viruliferous vector to land on infected plants | Varied (Default = 1) |
| $\omega_-$ | Probability that a non-viruliferous vector settles to feed on a susceptible plant | Varied (Default = 0.5) |
| $\omega_+$ | Probability that viruliferous vector settles to feed on a susceptible plant | Varied (Default = 0.5) |
| $\epsilon_-$ | Bias of non-viruliferous vector to settle to feed on an infected plant | Varied (Default = 1) |
| $\epsilon_+$ | Bias of a viruliferous vector to settle to feed on an infected plant | Varies (Default = 1) |
| $\phi_-$ | The average number of plants visited by a non-viruliferous vector per unit of time | Derived (Eq (11)) |
| $\phi_+$ | The average number of plants visited by a viruliferous vector per unit of time | Derived (Eq (13)) |
| $\gamma$ | Probability that an uninfected plant is inoculated by a single visit from a viruliferous vector | Eq (7) |
| $\eta$ | Probability that a non-viruliferous vector acquires the virus in a single visit to an infected plant | Eq (8) |
| $\Lambda(S,I,Z)$ | Overall rate at which uninfected plants become infected | Derived |
| $\Omega(S,I,X)$ | Overall rate at which non-viruliferous vectors become viruliferous | Derived |
| $g(S,I,X,Z)$ | Rate at which vectors are born | Derived |
| $h_-(S,I)$ | Rate (per capita) at which non-viruliferous vectors die | Derived |
| $h_+(S,I)$ | Rate (per capita) at which viruliferous vectors die | Derived |
| $\kappa$ | Total density of vectors in the absence of disease | Derived |
| $R_0^{PV}$ | Average number of vectors directly acquiring virus by the introduction of a single infected plant | Derived |
| $R_0^{VP}$ | Average number of plants directly inoculated by the introduction of a single viruliferous vector | Derived |
| $R_0$ | Basic reproduction number | Derived $(R_0^2 = R_0^{PV} R_0^{VP})$ |

feed on uninfected vs. infected plants. In Eq (1), the composite parameters $\Lambda(S,I,Z)$ and $\Omega(S,I,X)$ control inoculation of healthy plants by viruliferous vectors and acquisition of the virus from infected plants to non-viruliferous vectors, respectively. In the absence of vector

preference, the forms we adopt would be

$$\Lambda(S, I, Z) = \phi\gamma Z \frac{S}{S+I},$$

$$\Omega(S, I, X) = \phi\eta X \frac{I}{S+I}.$$

(2)

The overall rate at which susceptible plants are inoculated, $\Lambda(S,I,Z)$, is therefore the product of: $\phi$, the average number of plants visited by each vector per unit of time; $\gamma$, the probability that an uninfected plant is inoculated with the virus during a single visit by a viruliferous vector; $Z$, the density of vectors that are currently viruliferous; and $S/(S+I)$, the probability that a single visit by a vector is made to an uninfected plant. Similarly, the overall rate at which non-viruliferous vectors acquire infection, $\Omega(S,I,X)$, depends on: $\phi$, the rate at which plants are visited; $\eta$, the probability that an non-viruliferous vector acquires the virus from an infected plant; $X$, the density of non-viruliferous vectors; and $I/(S+I)$, the probability that an individual visit is made to an infected plant.

## Vector landing preference

We extend the basic model by including vector preference. We define $v_-$ to be the degree to which non-viruliferous vectors prefer to land on infected plants. Non-viruliferous vectors (-) can then be biased either in favour of ($v_->1$), or against ($v_-<1$), landing on infected plants (*cf.* Fig 1)

$$P(\text{non-viruliferous vector lands on a susceptible plant}) = \frac{S}{S+v_-I},$$

$$P(\text{non-viruliferous vector lands on an infected plant}) = \frac{v_-I}{S+v_-I}.$$

(3)

Hence for $v_-\in[0,1)$ a non-viruliferous vector prefers to land on healthy, susceptible plants; for $v_-\in(1,\infty)$, it prefers to land on infected plants. For $v_- = 1$, there is no bias in landing. An analogous parameter $v_+$ controls the landing behaviour of viruliferous vectors (+), with

$$P(\text{viruliferous vector lands on a susceptible plant}) = \frac{S}{S+v_+I},$$

$$P(\text{viruliferous vector lands on an infected plant}) = \frac{v_+I}{S+v_+I}.$$

(4)

## Vector feeding preference

We assume that an individual vector probes a plant directly after landing, but then chooses between settling for an extended feed or immediately moving off to probe a different plant. The probability of feeding potentially depends on both the state of the vector (non-viruliferous or viruliferous) and of the plant (healthy or infected). The parameters $\omega_-$ and $\omega_+$ are the probabilities that non-viruliferous (-) or viruliferous (+) vectors settle to feed on healthy susceptible plants. The parameters $\epsilon_-$ and $\epsilon_+$ then control any bias for ($\epsilon_\pm>1$) or against ($\epsilon_\pm<1$) settling on infected plants. For non-viruliferous vectors the probabilities of settling and feeding vs. probing and moving elsewhere are therefore

$$P(\text{non-viruliferous vector settles for extended feed on a susceptible plant}) = \omega_-,$$

$$P(\text{non-viruliferous vector probes a susceptible plant but moves elsewhere}) = 1-\omega_-,$$

$$P(\text{non-viruliferous vector settles for extended feed on an infected plant}) = \varepsilon_-\omega_-,$$

$$P(\text{non-viruliferous vector probes an infected plant but moves elsewhere}) = 1-\varepsilon_-\omega_-.$$

(5)

The corresponding probabilities for viruliferous vectors are

$$P(\text{viruliferous vector settles for extended feed on a susceptible plant}) = \omega_+,$$
$$P(\text{viruliferous vector probes a susceptible plant but moves elsewhere}) = 1-\omega_+,$$
$$P(\text{viruliferous vector settles for extended feed on an infected plant}) = \varepsilon_+\omega_+,$$
$$P(\text{viruliferous vector probes an infected plant but moves elsewhere}) = 1-\varepsilon_+\omega_+.$$

(6)

As well as the obvious constraint that $\omega_\pm \leq 1$, we note that parameters must also be constrained such that $\epsilon_\pm\omega_\pm \leq 1$ in order for all probabilities in Eqs (5) and (6) to be well-posed.

## Effect of transmission type on the probability of transmission per visit

We account for the different effects of probing vs. feeding on the spread of viruses that are non-persistently transmitted (NPT) vs. persistently transmitted (PT) [34]. We assume the probability that uninfected plants are inoculated with the virus during a single visit by a viruliferous vector ($\gamma$) depends on the fraction of visits in which it probes or feeds, with

$$\gamma = \begin{cases} 1 & \text{for NPT viruses} \\ \omega_+ & \text{for PT viruses} \end{cases},$$

(7)

since PT viruses can only be inoculated by viruliferous vectors if they settle for extended feeds. For NPT viruses we assume that a single visit is sufficient for inoculation.

However, the probability of acquisition by a non-viruliferous vector during a single visit to an infected plant is given by

$$\eta = \begin{cases} 1 - \varepsilon_-\omega_- & \text{for NPT viruses} \\ \varepsilon_-\omega_- & \text{for PT viruses} \end{cases},$$

(8)

because virions of NPT viruses are detached from the vector's stylet back into the same plant during an extended feed, whereas extended feeding is required to acquire PT viruses [34].

## Number of plants visited per vector per unit of time

The distinction between probing and feeding will affect the length of the average visit to a plant, which in turn affects the overall rate at which plants are visited by vectors. We assume the duration of an extended feed by a given vector is fixed at $\Gamma$, and that probing is instantaneous compared to feeding. Defining $\Delta_{--}$ and $\Delta_{-+}$ as the average lengths of visits by non-viruliferous vectors to uninfected and infected plants, respectively

$$\Delta_{--} = \omega_- \times \Gamma + (1 - \omega_-) \times 0 = \omega_-\Gamma,$$
$$\Delta_{-+} = \varepsilon_-\omega_- \times \Gamma + (1 - \varepsilon_-\omega_-) \times 0 = \omega_-\varepsilon_-\Gamma.$$

(9)

Given the probabilities that non-viruliferous vectors land on uninfected vs. infected plants, the length of the average visit by a non-viruliferous vector is therefore

$$\Delta_- = \frac{S}{S + v\_I} \times \omega_-\Gamma + \frac{v\_I}{S + v\_I} \times \omega_-\varepsilon_-\Gamma = \frac{\omega_-\Gamma(S + v\_\varepsilon\_I)}{S + v\_I}.$$

(10)

If we furthermore assume the time spent by vectors in moving between pairs of plants is very small compared to $\Gamma$ (where we note in support of this assumption that in our default parameterization $\Gamma$ is taken to be of the order of days), then $\phi_-$, the average number of plants

visited by a non-viruliferous vector per unit of time, is

$$\phi_- = \frac{1}{\Delta_-} = \frac{S + v_- I}{\omega_- \Gamma(S + v_- \varepsilon_- I)}. \tag{11}$$

An analogous calculation shows the length of an average visit by a viruliferous vector is

$$\Delta_+ = \frac{S}{S + v_+ I} \times \omega_+ \Gamma + \frac{v_+ I}{S + v_+ I} \times \omega_+ \varepsilon_+ \Gamma = \frac{\omega_+ \Gamma(S + v_+ \varepsilon_+ I)}{S + v_+ I}, \tag{12}$$

and the average number of plants visited by a viruliferous vector per unit of time, $\phi_+$, is

$$\phi_+ = \frac{1}{\Delta_+} = \frac{S + v_+ I}{\omega_+ \Gamma(S + v_+ \varepsilon_+ I)}. \tag{13}$$

## Vector population dynamics

Plant host-virus-vector models often consider only the simplified situation in which the population size of the vector was fixed, with births/deaths and immigration/emigration always held balanced by density-dependent factors. However, this assumption makes it impossible to consider any effects on epidemics of conditional vector preference that depend on vector population dynamics [48]. We therefore include births and deaths of vectors explicitly in the model.

We assume a general form of logistic growth controls the net birth rate of vectors

$$g(S, I, X, Z) = (\sigma_- X + \sigma_+ Z)\left(1 - \frac{X + Z}{\zeta}\right), \tag{14}$$

in which $\sigma_-$ is the per capita rate of reproduction of non-viruliferous vectors when there is no density dependence, $\sigma_+$ is the same rate for viruliferous vectors, and $\zeta$ is the density of the vector population at which the birth rate is zero. The assumption is made that the maximum density of the vector population $\zeta$ does not depend on the infection status of the reproducing vectors; instead, we assume the birth rate depends on the infection status of the plants on which vectors feed.

We denote the fraction of feeds on infected plants by $f_-(S,I)$ for non-viruliferous vectors, and $f_+(S,I)$ for viruliferous vectors, where both quantities depend on the proportions of infected vs. uninfected plants in the host population. We assume the maximum rates of reproduction are given by

$$\begin{aligned}
\sigma_- &= \sigma(1 + (\beta - 1)f_-(S, I)), \\
\sigma_+ &= \sigma(1 + (\beta - 1)f_+(S, I)),
\end{aligned} \tag{15}$$

where $\sigma$ is the maximum rate for vectors that feed entirely on uninfected plants and $\sigma\beta$ is the corresponding maximum rate if only infected plants were fed upon (i.e. when $f_\pm = 1$). The parameter $\beta$ may therefore be interpreted as the relative reproduction rate on infected plants, and so in general values $\beta < 1$ and $\beta > 1$ are possible (as well as the default assumption, $\beta = 1$).

Specification of the model of the vector birth rate is completed by calculating $f_\pm(S,I)$. For non-viruliferous vectors, combining Eqs (3) and (5) indicates

$$\begin{aligned}
p_{--} &= P(\text{next visit leads to an extended feed on a susceptible plant}) = \frac{\omega_- S}{S + v_- I}, \\
p_{-+} &= P(\text{next visit leads to an extended feed on an infected plant}) = \frac{\omega_- \varepsilon_- v_- I}{S + v_- I}.
\end{aligned} \tag{16}$$

The fraction of feeds by non-viruliferous vectors that are on infected plants is therefore

$$f_-(S, I) = \frac{p_{-+}}{p_{--} + p_{-+}} = \frac{\omega_- \varepsilon_- v_- I}{\omega_- S + v_- \omega_- \varepsilon_- I} = \frac{v_- \varepsilon_- I}{S + v_- \varepsilon_- I}. \tag{17}$$

The corresponding quantity for viruliferous vectors is

$$f_+(S, I) = \frac{v_+ \varepsilon_+ I}{S + v_+ \varepsilon_+ I}. \tag{18}$$

The full model of the birth rate of vectors is therefore given by

$$g(S, I, X, Z) = \sigma\left( (X + Z) + (\beta - 1)I\left( \frac{v_- \varepsilon_- X}{S + v_- \varepsilon_- I} + \frac{v_+ \varepsilon_+ Z}{S + v_+ \varepsilon_+ I} \right) \right)\left( 1 - \frac{X + Z}{\zeta} \right), \tag{19}$$

where $\sigma$ is the maximum per capita rate of reproduction for vectors that feed entirely on uninfected plants, $\beta$ is the relative rate of vector reproduction on infected plants, and $\zeta$ is the size of the vector population at which the birth rate is zero.

To model an overall rate of loss of vectors from the system, we assume the per capita rate of death of vectors is also affected by the number of plants visited per vector per extended feed. Moving between plants is energy consuming and puts the vector at additional risk, e.g. of predation. Losses also potentially occur because vectors making flights may simply move away from the plant population, especially non-colonising vectors. The same argument as was used to derive Eq (11) indicates that for non-viruliferous vectors

$$\theta_- = \text{Average number of plants visited by a non-viruliferous vector per feed,}$$
$$= \frac{S + v_- I}{\omega_-(S + v_- \varepsilon_- I)} = \Gamma\phi_-. \tag{20}$$

The analogous quantity for viruliferous vectors is

$$\theta_+ = \text{Average number of plants visited by a viruliferous vector per feed,}$$
$$= \frac{S + v_+ I}{\omega_+(S + v_+ \varepsilon_+ I)} = \Gamma\phi_+. \tag{21}$$

We assume that the per capita vector death rate can include a component that increases linearly with the number of plants visited per feed, taking

$$h_-(S, I) = \alpha(1 + \delta(\theta_- - 1)) = \alpha(1 + \delta(\Gamma\phi_- - 1)),$$
$$h_+(S, I) = \alpha(1 + \delta(\theta_+ - 1)) = \alpha(1 + \delta(\Gamma\phi_+ - 1)). \tag{22}$$

In Eq (22) $\alpha$ is the underlying rate at which vectors die, and $\delta$ controls any increase in the death rate as more plants are visited per feed. Since the number of plants visited per feed is constrained via $\theta_\pm \geq 1$, the form we have taken corresponds to an additional death rate for all values $\delta > 0$.

The vector population in the absence of disease is a derived quantity in our model. Denoting this population size by $\kappa$, it follows by seeking an equilibrium value of the model when $I = 0$ that

$$0 = g(N, 0, \kappa, 0) - h_-(N, 0)\kappa = \kappa\left[ \sigma\left( 1 - \frac{\kappa}{\zeta} \right) - \alpha\left( 1 + \delta\left( \frac{1}{\omega_-} - 1 \right) \right) \right], \tag{23}$$

meaning the vector carrying capacity when the pathogen is absent is given by

$$\kappa = \zeta\left(1 - \frac{\alpha}{\sigma}\left(1 + \delta\left(\frac{1}{\omega_-} - 1\right)\right)\right). \tag{24}$$

The vector population can only be self-sustaining in the absence of disease (i.e. $\kappa > 0$) if the maximum birth rate on uninfected plants is sufficiently large, i.e. when

$$\sigma > \alpha\left(1 + \delta\left(\frac{1}{\omega_-} - 1\right)\right). \tag{25}$$

## Summary of the model

The model is

$$\begin{aligned}
\frac{dS}{dt} &= \rho N - \Lambda(S, I, Z) - \rho S, \\
\frac{dI}{dt} &= \Lambda(S, I, Z) - (\rho + \mu)I, \\
\frac{dX}{dt} &= g(S, I, X, Z) + \tau Z - \Omega(S, I, X) - h_-(S, I)X, \\
\frac{dZ}{dt} &= \Omega(S, I, X) - \tau Z - h_+(S, I)Z,
\end{aligned} \tag{26}$$

in which transmission is controlled by

$$\Lambda(S, I, Z) = \phi_+ \gamma Z \frac{S}{\omega_+ \Gamma(S + v_+\varepsilon_+ I)}, \qquad \gamma = \begin{cases} 1 & \text{for NPT viruses} \\ \omega_+ & \text{for PT viruses} \end{cases}, \tag{27}$$

$$\Omega(S, I, X) = \phi_- \eta X \frac{v_- I}{\omega_- \Gamma(S + v_-\varepsilon_- I)}, \qquad \eta = \begin{cases} 1 - \varepsilon_-\omega_- & \text{for NPT viruses} \\ \varepsilon_-\omega_- & \text{for PT viruses} \end{cases},$$

and where for PT we allow the "plus" preference parameters to differ from the "minus" ones, whereas for NPT related pairs of parameters must be identical (i.e. parameters are constrained such that $v_- = v_+ = v$; $\omega_- = \omega_+ = \omega$ and $\epsilon_- = \epsilon_+ = \epsilon$).

The population dynamics of the vector depend on

$$\begin{aligned}
g(S, I, X, Z) &= \sigma\left((X + Z) + (\beta - 1)I\left(\frac{v_-\varepsilon_- X}{S + v_-\varepsilon_- I} + \frac{v_+\varepsilon_+ Z}{S + v_+\varepsilon_+ I}\right)\right)\left(1 - \frac{X + Z}{\zeta}\right), \\
h_-(S, I) &= \alpha(1 + \delta(\Gamma\phi_- - 1)), \\
h_+(S, I) &= \alpha(1 + \delta(\Gamma\phi_+ - 1)).
\end{aligned} \tag{28}$$

However, when $\beta = 1$ and $\delta = 0$ (default values), i.e. when vector preference does not interact with vector population dynamics, the functions controlling population dynamics become simpler, with

$$\begin{aligned}
g(S, I, X, Z) &= \sigma(X + Z)\left(1 - \frac{X + Z}{\zeta}\right), \\
h_-(S, I) &= \alpha, \\
h_+(S, I) &= \alpha.
\end{aligned} \tag{29}$$

### Model analysis and numerical approach

The invasion threshold and expressions determining the endemic equilibria can be derived analytically from Eq (26), with stability of equilibria found via the linearisation of the system in the vicinity of equilibrium. Qualitative and quantitative results are obtained analytically and numerically based on parameter ranges taken from the literature, with extensive numerical scans used to characterise the behaviour of the model, in a range of biologically relevant cases. Sensitivity analyses were performed relative to a pair of baseline parameterisations, one of which was chosen to exemplify non-persistent transmission, the other more appropriate for a persistently transmitted virus. As well as the differences between the model formulations to reflect systematic differences between the transmission classes (e.g. the ways in which values of $\gamma$ and $\eta$ in Eq (27) depend on model parameters), the key difference between scenarios turns on the numerical value of $\tau$, which controls the length of time for which an individual vector remains inoculative, i.e. able to cause virus infection of plants (see also Default Parameterisation in the Results section, below). Since the sensitivity analyses we perform are very extensive, our results therefore encompass scenarios relating to a wide range of host-virus-vector pathosystems.

### Online interface

An interactive online interface was created, allowing model behaviour for different parameterisations to be visualised. It is available at https://plantdiseasevectorpreference.herokuapp.com/ (Fig 2). This interface allows its user to explore the results of a single model run, or to perform a one-way parameter scan. In both cases, the user has the flexibility to choose parameterisation of the model, either using the default settings for PT or NPT, or by selecting a custom set of parameters. Whenever the user changes values of parameters, the interface triggers a callback function, which calculates the model outputs or equilibria before returning the results in the form of tables and interactive figures. The interface was built using Python Dash [58], which is open source. The source code for the interface is at https://github.com/nt409/vector-interface; this Python code also acts as a reference implementation of our mathematical model.

## Results

### Invasion threshold

We are interested in whether the virus can invade a compatible vector-plant system following first introduction and the extent to which vector preference contributes to successful invasion. The plant population density ($N$) in the absence of the virus is $S$ plants per unit area as all plants are healthy and susceptible. The vector population density ($\kappa$) in the absence of the virus is $X$ vectors per unit area as all vectors are non-viruliferous. We therefore assume that in the absence of virus the average population density of vectors per plant ($\kappa/N$) is at the carrying capacity per plant.

Suppose that when the virus is introduced there is a very small number of infected plants, $I \ll S \approx N$. From Eq (3), the probability a non-viruliferous vector lands on a susceptible plant is given by $S/(S+v_I)$, whereas the probability a non-viruliferous vector lands on an infected plant is $v_I/(S+v_I)$. Any infected plant would remain infectious for $1/(\mu+\rho)$ time units. The population density of non-viruliferous vectors is $\kappa$. From Eq (26), acquisition of virus from infected plants by the non-viruliferous vector population would therefore occur at rate

$$\phi_- \eta X \frac{v_I}{S + v_I} \approx \frac{\eta \kappa v_-}{\omega_- \Gamma N}, \tag{30}$$

per infected plant, since $\phi_- \approx 1/(\omega_- \Gamma)$ and $(v_I)/(S+v_I) \approx v_I/N$ when $I$ is small.

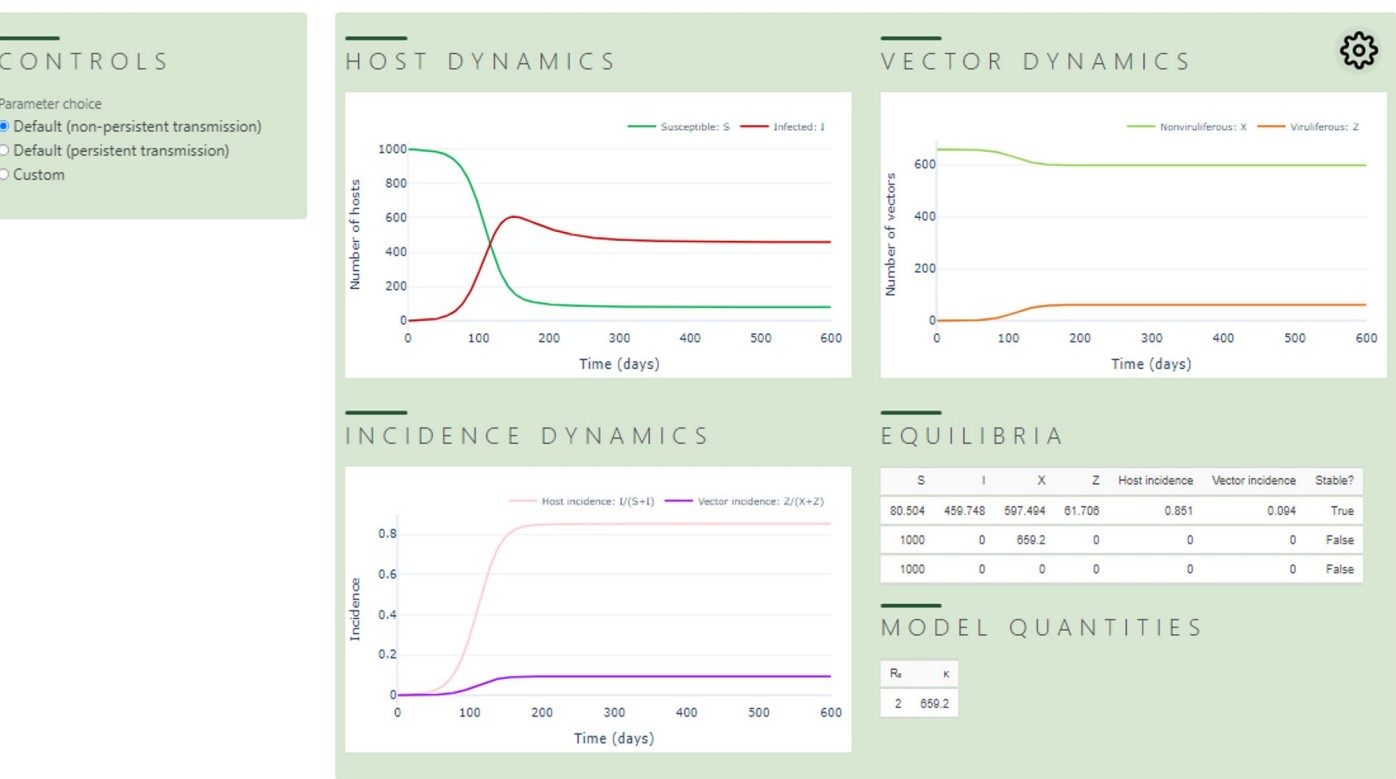

**Fig 2. Online interface to the model.** A screenshot showing the results of the "Model" tab of the interactive interface, available online at https://plantdiseasevectorpreference.herokuapp.com/. The user can select the model parameterisation to investigate (left hand side), and–by selecting between the "Model" and "Parameter Scan" tabs on the top-right–can choose to be presented with the dynamics of the model in time, or with a bifurcation diagram showing how the number, numeric value(s) and stability of equilibria depend upon the value of a model parameter. The cog-wheel icon at the top right of the main panel allows the user to select the number of columns used to display graphical results. The screenshot in the figure shows the results in "Two column" mode.

The average number of non-viruliferous vectors directly acquiring virus from a single infected plant over its infectious period is therefore

$$R_0^{PV} = \frac{\kappa}{N} \cdot \frac{\eta v_-}{\omega_- \Gamma (\mu + \rho)}.$$  (31)

Suppose for each newly viruliferous vector, at the time of introduction, $Z \ll X \approx \kappa$. From Eq (4), the probability a viruliferous vector lands on a susceptible plant is $S/(S+v_+I)$ and the probability a viruliferous vector lands on an infected plant is $v_+I/(S+v_+I)$. As the population density of infected plants ($I$) is small at the time of introduction, then from Eq (22), the per capita death rate of viruliferous vectors, $h_+(S,I)$ is approximately $\alpha(1+\delta(1/\omega_+ - 1))$. These vectors additionally lose infectivity at rate $\tau$. A viruliferous vector therefore remains inoculative for an average of

$$\frac{1}{\tau + \alpha \left(1 + \delta \left(\frac{1}{\omega_+} - 1\right)\right)}$$  (32)

time units. For each viruliferous vector in the plant population which, apart from the initially

introduced infected plant, is otherwise totally susceptible, inoculation occurs at rate

$$\phi_+ \gamma \frac{S}{S + v_+ I} \approx \frac{\gamma}{\omega_+ \Gamma}, \tag{33}$$

since $\phi_+ \approx 1/(\omega_+ \Gamma)$ and $S/(S+v_+ I) \approx 1$ when $I$ is small.

The average number of plants per unit area that would become directly infected for each viruliferous vector infected by the initially infected plant in an otherwise uninfected population is therefore

$$R_0^{VP} = \frac{\gamma}{\omega_+ \Gamma \left( \tau + \alpha \left( 1 + \delta \left( \frac{1}{\omega_+} - 1 \right) \right) \right)}. \tag{34}$$

We note that $v_+$ does not appear in this expression.

The pathogen's basic reproduction number is obtained by the product of $R_0^{VP}$ and $R_0^{PV}$, leading to

$$R_0^2 = \frac{\kappa}{N} \cdot \frac{\eta v_- \gamma}{\omega_- \omega_+ \Gamma^2 (\mu + \rho) \left( \tau + \alpha \left( 1 + \delta \left( \frac{1}{\omega_+} - 1 \right) \right) \right)}. \tag{35}$$

To ease interpretation, we back substitute $\phi_- \approx 1/(\omega_- \Gamma)$ and $\phi_+ \approx 1/(\omega_+ \Gamma)$, and rearrange the order of terms, leading to

$$R_0^2 = \frac{\kappa}{N} \cdot \frac{1}{\mu + \rho} \cdot \phi_- \eta v_- \cdot \frac{1}{\left( \tau + \alpha \left( 1 + \delta \left( \frac{1}{\omega_+} - 1 \right) \right) \right)} \cdot \phi_+ \gamma. \tag{36}$$

Successive terms in this expression allow an intuitive interpretation as follows:

> Average number of vectors per plant in absence of virus × average infectious period (time units) of a single infected plant × average number of plants visited by a vector (per unit time) × probability of virus acquisition by a single vector during a single visit × average period (time units) a vector remains viruliferous × average number of plants visited per vector (per unit time) × probability of inoculation by a single viruliferous vector.

The heuristic derivation of the basic reproduction number outlined above matches that can be obtained more formally from the spectral radius of the Next Generation Matrix (S2 Appendix). The squared formulation reflects the fact that there are two cycles involved in transmission, from plant to vector and from vector to plant [59]. Clearly, $R_0 \geq 0$ for non-negative values of the parameters (including $\kappa$, which can be positive or negative depending on the values of the other parameters; cf. Eq (24)). The virus invades the system whenever $R_0 > 1$, but note that the parameters $\gamma$ (Eq (7)), $\eta$ (Eq (8)) and $\kappa$ (Eq (24)), depend on other parameters and/or the transmission type (NPT or PT) of the virus in question. As noted above $v_-$ affects the numerical value of $R_0$, but $v_+$ does not. Any effect of $v_+$ depends on viruliferous vectors being attracted to infected plants, and during initial invasion this "second order" effect is small. However, the probabilities that a non-viruliferous ($\omega_-$) or a viruliferous ($\omega_+$) vector settles and feeds on a plant are both retained in the expression for $R_0$.

A basic reproduction number was derived from an earlier version of the current model in which a conditional vector preference parameter $v_Z$, consistent with $v_+$ here, appeared in the term $N + v_Z$ (Appendix A in [60]). However, that term is not valid dimensionally since $N$ is a population density (plants per unit area). When the appropriate corrections are made to the

past modelling work, the basic reproduction number is essentially the same as the derived $R_0^2$ here.

## Default parameterisation of the model

The default parameterisation (Table 1) assumes that infected plants are no better or worse than susceptible plants for vector reproduction ($\beta = 1$) and that there is no additional mortality associated with increased flights between plants ($\delta = 0$). The default density of vectors at which the birth rate becomes zero, $\zeta$, was selected–after fixing the other parameter values as described below–to ensure $R_0 = 2$ for both NPT and PT, giving $\Gamma = 2$ days and $\zeta = 1,977.6$ (NPT) and $\zeta = 163.2$ (PT). We note that with these changes $\kappa = 659.2$ (NPT) and $\kappa = 54.4$ (PT). The plant population density in the absence of disease was arbitrarily chosen to be $N = 1,000$ per unit area, noting that our approach of fixing $R_0$ to specify our parameterisation means that any alteration to this value would be scaled into values of other parameters. The vector population size is therefore between about 0.2 and 2 vectors/plant.

Other parameter values are informed, where possible, by previous modelling studies (e.g. [7,54–55, 61]). Parameters set in this way include the rate at which plants are harvested ($\rho = 0.01 \, d^{-1}$), the additional rate of disease-induced death and/or roguing ($\mu = 0.01 \, d^{-1}$), the underlying per capita vector death rate ($\alpha = 0.12 \, d^{-1}$), the maximum per capita vector birth rate ($\sigma = 0.18 \, d^{-1}$), and, assuming the rate at which vectors lose infectivity depends strongly on the transmission mode, $1/\tau = 0.25 \, d$ (NPT) and $1/\tau = 20 \, d$ (PT). There is no direct analogue of $\Gamma$, the length of a feed, in previous modelling work. We therefore arbitrarily assume a value of 2 d, irrespective of the transmission type, to set the number of plants visited per unit time in the model. Assuming, at least by default, that there is no vector preference for landing based on infection status, we take the values $\nu_- = \nu_+ = 1$, and for feeding we take $\omega_- = \omega_+ = 0.5$, with $\epsilon_- = \epsilon_+ = 1$. By default, therefore, individual vectors move between plants once every two days, and on average probe and move away from one plant before settling on the second plant they visit.

We note, however, that–as well as the extensive sensitivity scans that are described below–the "Custom" parameterisation option in our online interface (Fig 2) allows all parameters to be freely set. This allows the reader of our paper to investigate, in some detail, the behaviour of the model for any parameterisation.

## Time plots for baseline parameterisation

Densities per unit area of healthy susceptible plants, infected plants, non-viruliferous vectors, and viruliferous vectors, following the introduction of one infected plant per unit area into an otherwise healthy plant population and one viruliferous vector per unit area into an otherwise non-viruliferous vector population, are shown in Fig 3. The models were run for 600 days to ensure equilibrium, but this was approached in less than 200 days for NPT and a little more than 200 days for PT. The density of infected hosts and disease incidence in the plant population was always higher with NPT, with a much lower density of viruliferous vectors than with PT. The proportion of viruliferous vectors in the vector populations was much higher with PT.

## Sensitivity to life history and epidemiological parameters

The sensitivities of $R_0$, the equilibrium density of infected hosts $I_\infty$, and the equilibrium disease incidence $I_\infty/(S_\infty + I_\infty)$ to changes in model parameters are shown in Fig 4. In these plots, the vector preference parameters for landing and feeding are set to 1; hence sensitivity to the derived inoculation $\gamma$ and acquisition $\eta$ probabilities are not shown. For $R_0$, the responses to vector death rate ($\alpha$) show the greatest divergence when NPT (Fig 4A) and PT (Fig 4D) are

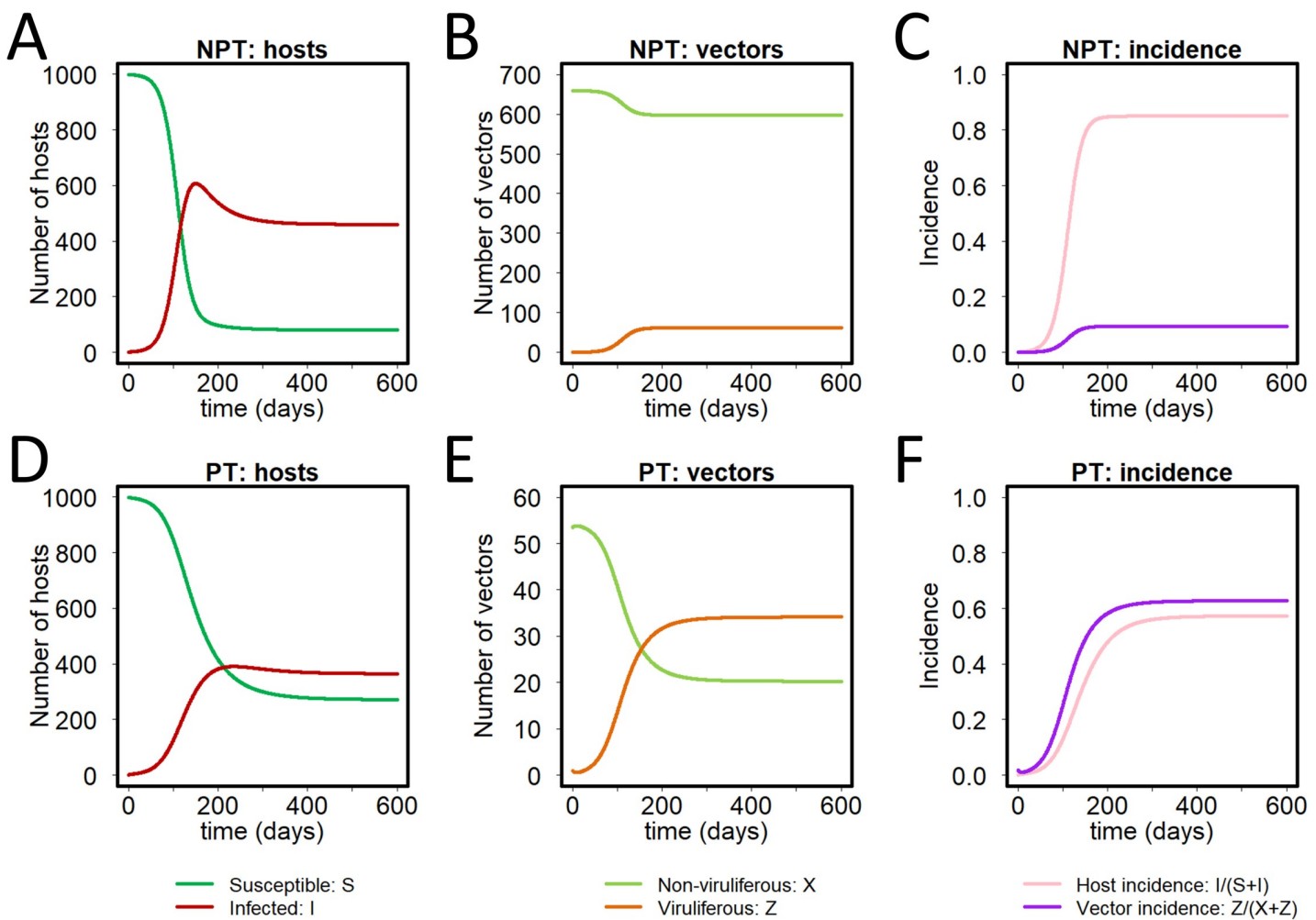

**Fig 3. Baseline parameterisation of the model.** Models were parameterised using available literature, selecting parameters to ensure a default value of $R_0 = 2$ for both the NPT and PT models. **(A)** Numbers of susceptible and infected hosts; **(B)** numbers of non-viruliferous and viruliferous vectors; and **(C)** incidences (i.e. proportions infected) for both hosts and vectors, as functions of time, using the default parameterisation of the NPT model. **(D)-(F)** The same for the PT model. All disease progress curves are shown for the case in which a single infected host and viruliferous vector are simultaneously introduced at time $t = 0$, and the models were run for 600 days (a value selected to be sufficiently large for equilibrium to be attained for both parameterisations).

compared. The $R_0$ value is higher for PT than NPT at low death rates. The sensitivity to other parameters is similar for both transmission types, especially when parameters are varied above the default values. For $I_\infty$, the greatest divergence between NPT and PT occurs for the host roguing rate ($\mu$), with higher values at lower rates for NPT. There are also differences in responses to the initial plant population size ($N$), with an almost linear increase in $I_\infty$ as $N$ increases above the default for NPT, whereas it levels off for PT. The equilibrium disease incidences for NPT and PT largely reflect the fact that, at least for our default model parameterisation, overall disease incidence was higher for NPT (cf. Fig 3).

## Model equilibria

There are two disease-free equilibria, $(S,I,X,Z) = (N,0,0,0)$, and $(S,I,X,Z) = (N,0,\kappa,0)$. If the parameters are such that $\kappa < 0$ then the equilibrium in which $X = 0$ is always stable. If $\kappa > 0$, and so if the vector population can persist in the absence of disease, then the equilibrium in which

$X = 0$ is always unstable. The stability of the equilibrium in which $X = \kappa > 0$ is controlled by the basic reproduction number: if $R_0 \leq 1$ then this disease-free equilibrium is stable, otherwise it is unstable (S3 Appendix).

In general the model has up to four non-zero equilibria in which $I \neq 0$. Often only certain of these equilibria can ever be attained in practice, since the other equilibria do not necessarily take biologically-meaningful values in which all state variables positive at equilibrium. The equilibrium values of $I$ can be calculated explicitly via the solutions of a quartic equation, the coefficients of which can be determined by a routine–albeit very long winded–calculation (S3 Appendix). The values of all other state variables at equilibrium can then be determined as functions of these values of $I$.

In the case when $\beta = 1$ and $\delta = 0$, i.e. when vector preference does not interact with vector population dynamics, further simplification is possible (S3 Appendix). In this case it can be shown that the equilibrium values of $I$ are given by solutions to a certain quadratic equation. In this case there is therefore a maximum of two non-zero equilibria, although again it is possible that one or both equilibria do not take biologically meaningful values.

For both of our default parameterisations of the model, there is a total of three biologically meaningful equilibria (the two disease-absent equilibria, as well as a single equilibrium with $I \neq 0$; the other disease-present equilibrium from the relevant quadratic has at least one negative state variable). Since $\kappa > 0$, only the disease-present equilibrium is locally stable, and so the long-term behaviour of the model is unambiguous and does not depend on initial conditions.

However, it is clearly of interest to understand whether multiple disease-present equilibria can be attained, and if so, whether they are stable or not. To assess this, we did an extensive randomisation scan (S4 Appendix), finding the number and stability of equilibria as parameters were randomly varied. Bistability between one of the vector-present/disease-absent or the vector-absent/disease-absent equilibrium and a single disease-present equilibrium was relatively commonplace (around 25–30% of cases). There was also a very small proportion ($\ll 1\%$ of cases) with bistability between two disease-present equilibria (with or without an additional stable disease-absent equilibrium); examples are given in S4 Appendix. In the remainder of the results as presented in the main paper, parameters were such that only the simpler bistability involving a single disease-present equilibrium was seen.

## Bistability

A particular example showing the effect of bistability between the disease-present and vector-present/disease-absent equilibrium was investigated in some detail (Fig 5), focusing upon the effects of $\mu$, the host roguing/additional disease-induced mortality parameter. Note that for both NPT and PT, the basic reproduction number is a decreasing function of $\mu$, which means $R_0 < 1$ for large values of $\mu$ (Fig 5A and 5D). For both models, at the default parameterisation ($\mu = 0.01 \; d^{-1}$; shown by a dot) there is a single biologically meaningful disease-present equilibrium, as well as the disease-free equilibrium (at which the vector is present since $\kappa > 0$). The other equilibrium for which $I \neq 0$ has a negative value of $I$. As $\mu$ increases, for both NPT and PT, there is a transcritical bifurcation at $\mu \approx 0.07 \; d^{-1}$ (i.e. at the threshold value $R_0 = 1$). This leads to an exchange of stability between the negative equilibrium–which becomes positive as $\mu$ is further increased–and the disease-absent equilibrium, meaning that for larger values of $\mu$ both the other disease-present equilibrium and the disease-absent equilibria are stable. The model then exhibits bistability; the equilibrium that is attained in the long term depends on the initial condition, with different basins of attraction (Fig 5B and 5E) leading to different results in the long term (Fig 5C and 5F). The only qualitative difference between the results for the two parameterisations is that, for PT, only the disease-free equilibrium exists for very large

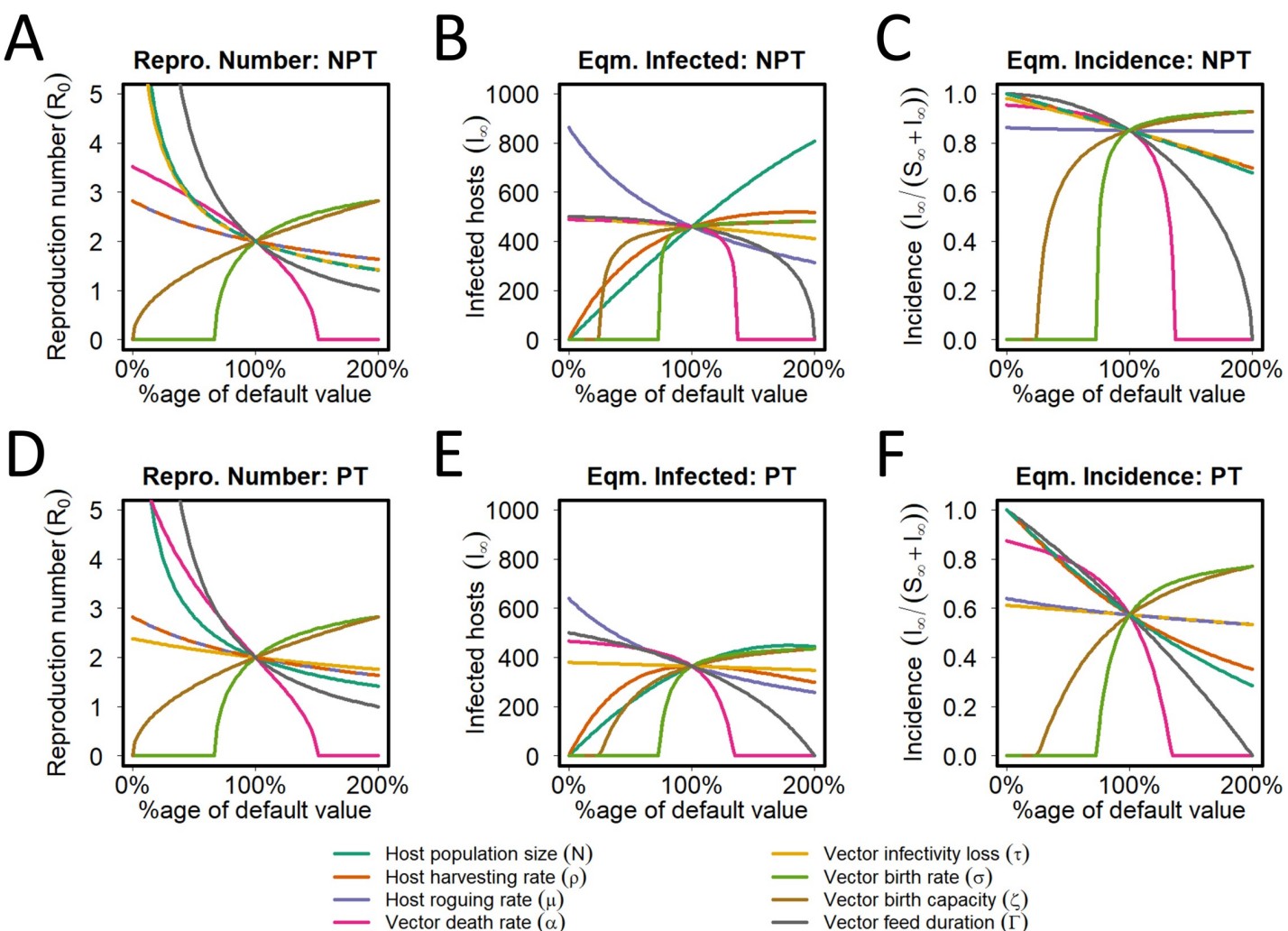

**Fig 4. Sensitivity of underlying model to life history and epidemiological parameters.** Responses of: **(A)** the basic reproduction number, $R_0$; **(B)** the equilibrium number of infected hosts, $I_\infty$; and **(C)** the equilibrium incidence, $I_\infty/(S_\infty+I_\infty)$, as the life history and epidemiological parameters in the model were varied between 0% and 200% of their default values for the NPT model. **(D)-(F)** The same for the PT model.

roguing rates. In particular, there is a saddle node bifurcation at $\mu\approx0.1868\ d^{-1}$, in which the stable and unstable disease-present equilibria collide and annihilate each other, meaning that for sufficiently large values of $\mu$ only the disease-free equilibrium exists. This actually also occurs for NPT, but at a much larger value of $\mu$, well outside the range of values shown in Fig 5.

## Vector preference

The expression for $R_0$ (Eq (36)) indicates invasion for both NPT and PT viruses is promoted by increasing $v_->1$, since it increases the probability of non-viruliferous vectors landing on infected plants, and so the number of plant-vector contacts that could lead to virus acquisition. Increasing $v_+$, the bias of viruliferous vectors for infected plants, would effectively be "wasting" contacts as when the virus is rare, contacts between infected plants and viruliferous vectors are very unlikely. Therefore, the value of $v_+$ does not affect invasion. Increasing $\epsilon_+$, thereby making viruliferous vectors more likely to feed on infected plants, also has no effect on invasion, for the same reason.

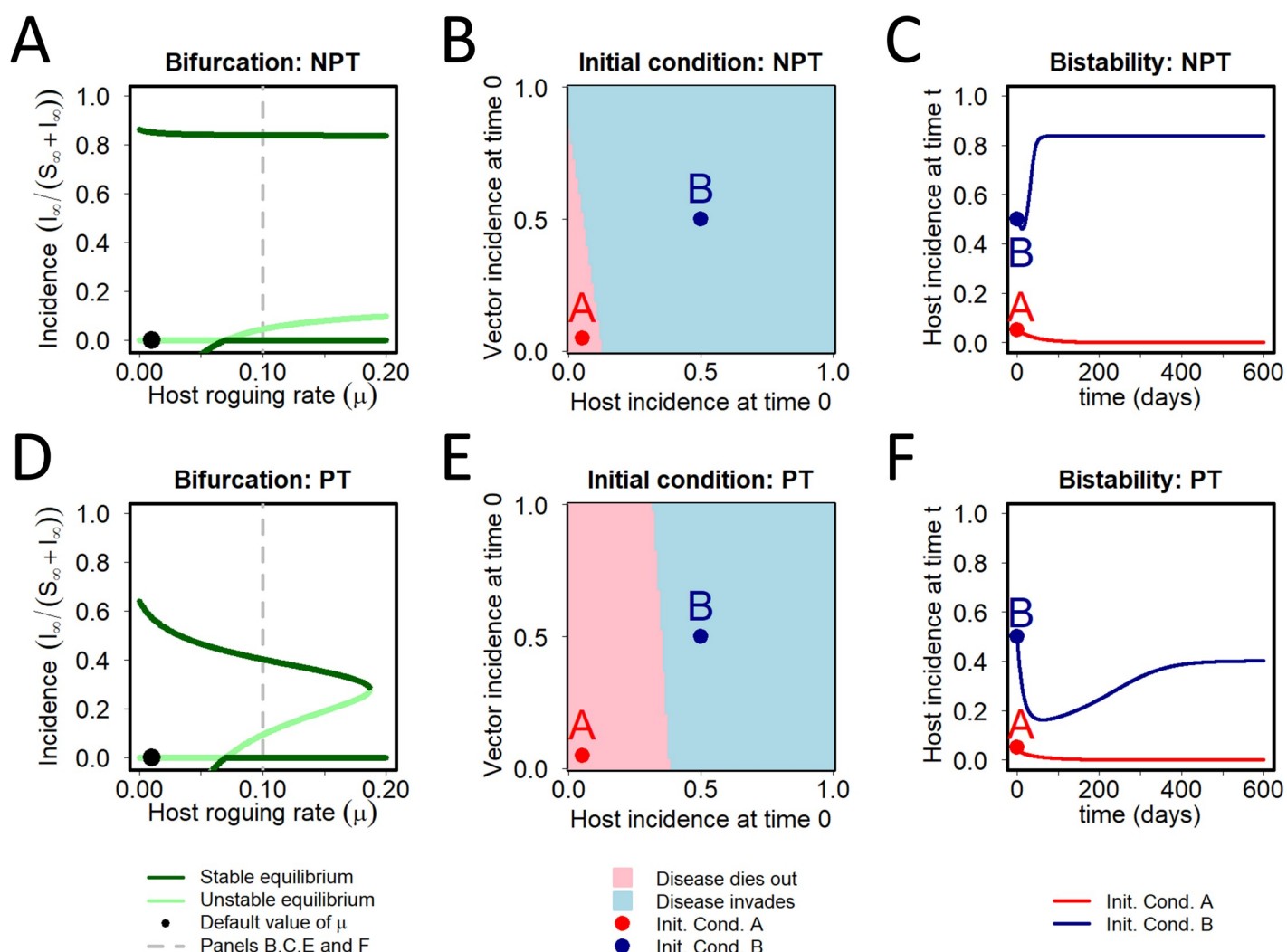

**Fig 5. Bistability.** For certain sets of parameters, the model has multiple stable biologically plausible equilibria; we illustrate this here by considering how the behaviour of the model depends upon the rate at which hosts are rogued. **(A)** A bifurcation diagram for the NPT model, using the host roguing rate ($\mu$) as the bifurcation parameter, and showing host disease incidence at equilibrium on the y-axis. When $\mu = 0.07$ d$^{-1}$ (i.e. seven times larger than the default value of 0.01 d$^{-1}$) the basic reproduction number $R_0 = 1$. Because $R_0$ is a decreasing function of the roguing rate (Eq (36)), it follows that $R_0 < 1$ for larger values of $\mu$, and so the disease-free equilibrium becomes locally stable. However, there is a locally stable equilibrium corresponding to disease being present in the system is still present (as well as a further unstable equilibrium). **(B)** When $\mu = 0.1$ d$^{-1}$, which of the NPT model's two locally stable equilibria–the disease-free equilibrium or the disease-present equilibrium at which the incidence is around 0.8 –is eventually attained depends on the initial conditions. Qualitatively similar results occur for all values of $\mu > 0.07$ d$^{-1}$. **(C)** Host disease incidence in the NPT model as a function of time starting at initial conditions marked A ($I_0/(S_0+I_0)$, $Z_0/(X_0+Z_0)) = (0.05,0.05)$ and B ($I_0/(S_0+I_0)$, $Z_0/(X_0+Z_0)) = (0.5,0.5)$ in (B), with all other parameters equal to the defaults (apart from $\mu = 0.1$ d$^{-1}$). **(D)**-**(F)** The same for the PT model. The only qualitative difference between the results of the two versions of the model is that, in the PT model, only the disease-free equilibrium exists for very large roguing rates (there is in fact a saddle node bifurcation at $\mu \sim 0.1868$ d$^{-1}$ in which the stable and unstable disease-present equilibria collide and annihilate each other).

However, the vector preference parameters, $\omega_-$, $\omega_+$ and $\epsilon_-$ have opposite effects on invasion depending on the transmission type, NPT or PT. Since non-viruliferous vectors must feed on infected plants to acquire PT viruses, increasing $\epsilon_-$ promotes invasion in this case. However, invasion of NPT viruses would be reduced, since feeding stops the virus being acquired by the vector as virus is lost from the stylet [34]. Increasing the underlying probability of feeding by either uninfected ($\omega_-$) or infected ($\omega_+$) vectors is also deleterious for NPT, essentially because feeding "wastes" time. However, for PT viruses, feeding is required for both acquisition and inoculation.

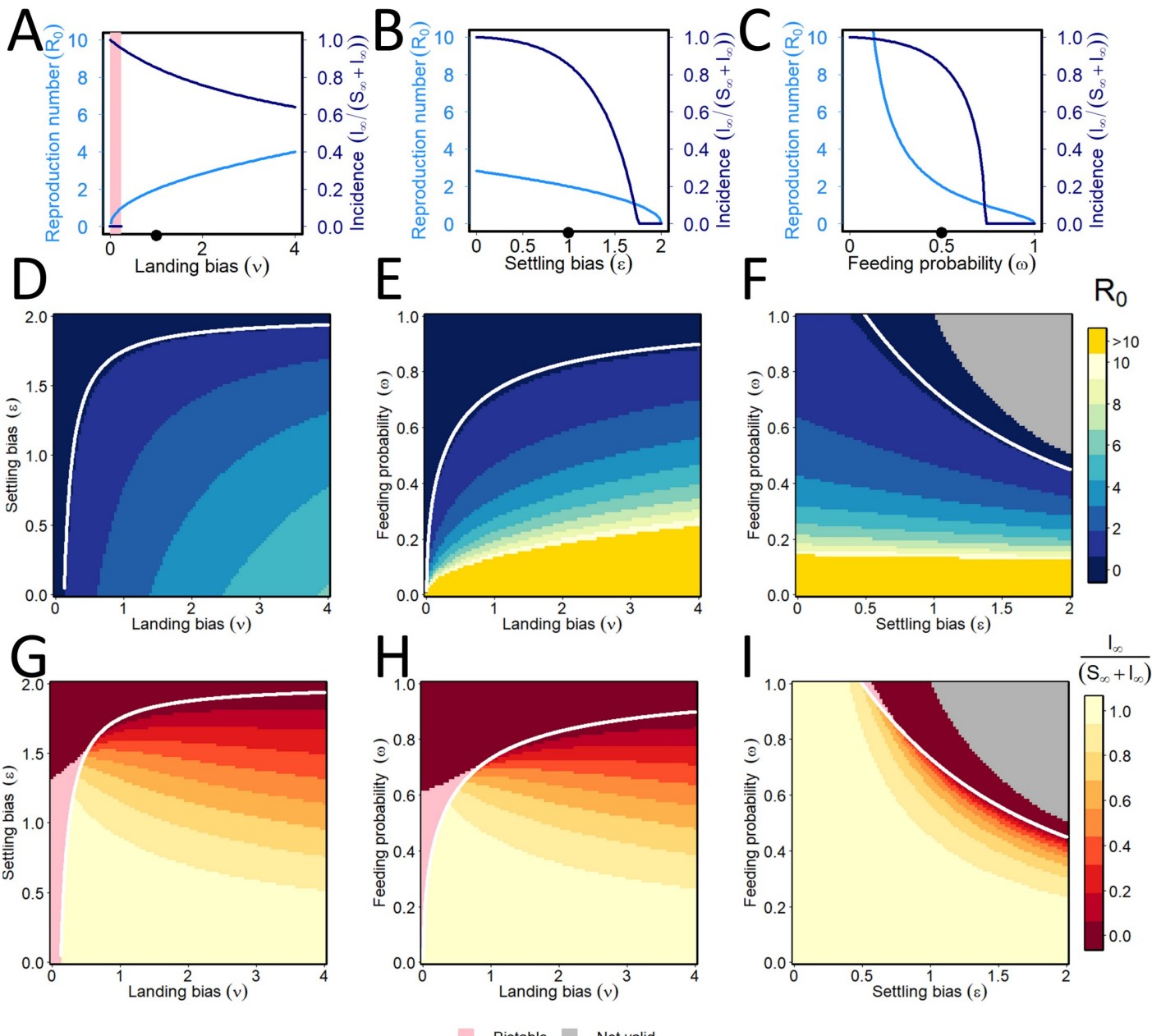

**Fig 6. Non-persistent transmission: effect of vector preference. (A)-(C)** Responses of the basic reproduction number (light blue) and equilibrium disease incidence (dark blue) to changes in the vector preference parameters: (A) landing bias ($\nu$); (B) settling bias ($\epsilon$) (C) feeding probability ($\omega$). For NPT there can be no conditional preference, so we introduce $\nu = \nu_- = \nu_+$, $\epsilon = \epsilon_- = \epsilon_+$ and $\omega = \omega_- = \omega_+$. Default values $\nu = \epsilon = 1$, $\omega = 0.5$ are marked with black dots on each plot's x-axis. As the parameter $\nu$ is altered in a one-way sensitivity analysis around the default values of other parameters, the model exhibits bistability whenever $R_0 < 1$. Therefore, which of the stable disease-free equilibrium and the stable disease-present equilibrium is attained in the long-term depends on the initial conditions. The relevant range of the parameter $\nu$ is marked by pink shading. **(D)-(I)** Responses of the basic reproduction number (top row: (D)-(F)) and the equilibrium incidence (bottom row: (G)-(I)) as pairs of vector preference parameters are simultaneously altered. Pairs of parameters for which $R_0 = 1$ are marked with a white curve. The grey regions in panels (F) and (I) correspond to invalid combinations of parameters for which $\omega\epsilon > 1$. Pairs of parameters for which the model exhibits bistability are again marked in pink. In panels (E) and (F) values of $R_0 > 10$ are shown in orange so as not to skew the colour-ramp and so obscure the main features of the results.

In our numerical work, we first considered vector preference for NPT viruses (Fig 6), for which there is guaranteed to be no conditional preference based on the vector's infective status, i.e. in the model $\nu = \nu_- = \nu_+$, $\epsilon = \epsilon_- = \epsilon_+$ and $\omega = \omega_- = \omega_+$. For NPT, this is assumed to always be

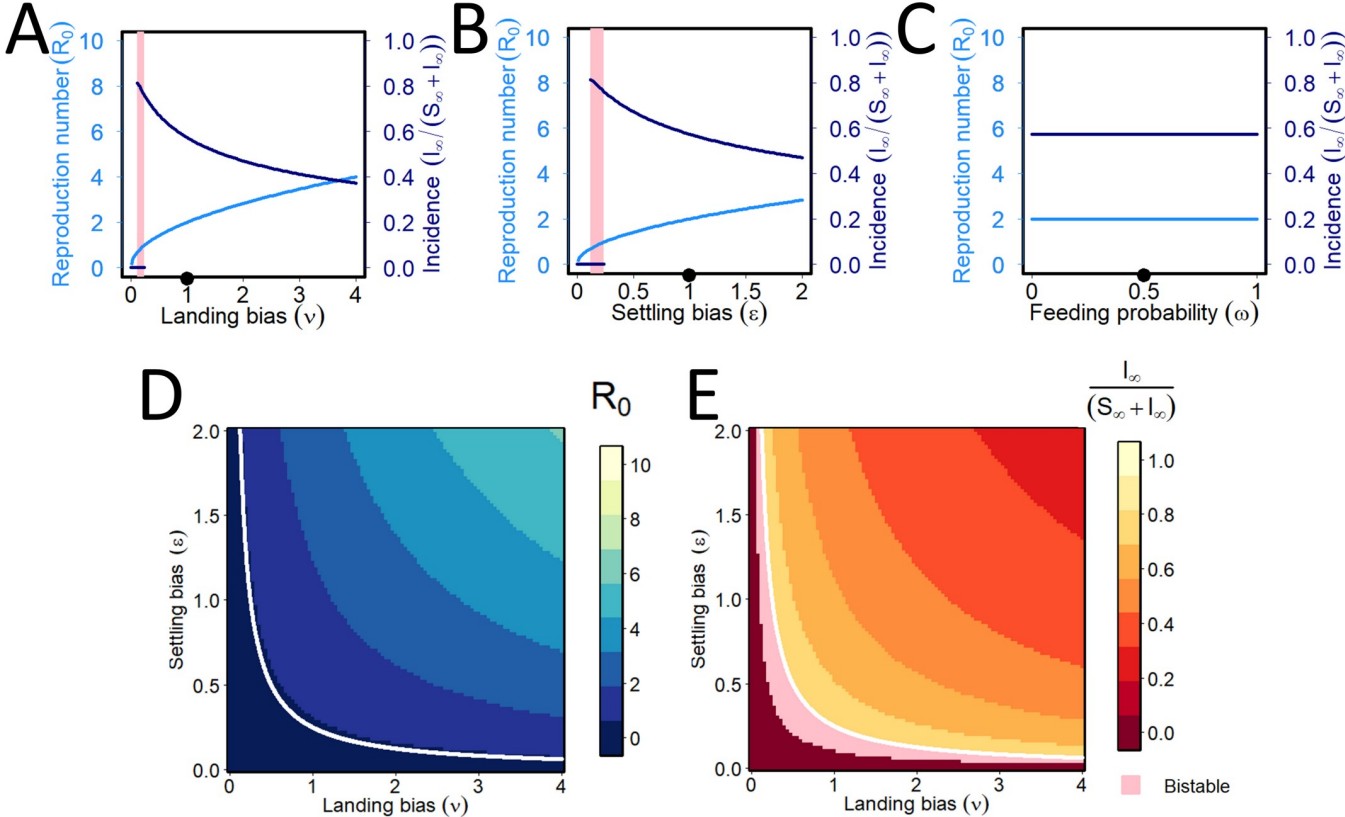

**Fig 7. Persistent transmission: effect of vector preference. (A)-(C)** Responses of the basic reproduction number (light blue) and equilibrium incidence (dark blue) to changes in the vector preference parameters: (A) landing bias ($v$); (B) settling bias ($\epsilon$) (C) feeding probability ($\omega$). For PT, conditional vector preference is biologically meaningful (shown in Fig 8), but in its absence, the model exhibits bistability in one-way sensitivity analyses to both the landing bias ($v$) and the settling bias ($\epsilon$) parameters (panels A and B). There is no sensitivity in response to feeding probability (panel C). In contrast to NPT, the model is not bistable whenever $R_0<1$ for a one-way sensitivity analysis altering $v$ (or $\epsilon$). Instead, there are saddle node bifurcations in which the stable and the unstable disease-present equilibria collide and annihilate each other, at around $v = \epsilon = 0.1016$ (note the unstable equilibria are not shown on this Figure, so the bifurcation is represented by the disease-present equilibrium abruptly "stopping" as the parameter values get smaller). **(D)-(E)** Responses of the basic reproduction number (D) and the equilibrium incidence (E) as landing bias and settling bias are simultaneously altered. Pairs of parameters for which $R_0 = 1$ are again marked with a white curve. Pairs of parameters for which the model exhibits bistability are again marked in pink.

the case due to the transient nature of virus acquisition, retention, and inoculation. For increasing values of the landing bias $v$, the basic reproduction number, $R_0$, increases while final disease incidence, $I_\infty/(S_\infty+I_\infty)$, decreases (Fig 6A). With increasing values of the settling bias $\epsilon$, $R_0$ decreases and final disease incidence drops considerably beyond the default value (Fig 6B), an effect even more pronounced with feeding probability, $\omega$ (Fig 6C). As the parameter $v$ decreases from the default value, the model exhibits bistability for $R_0<1$ (shown in pink). As the settling bias and feeding probability for NPT increase for any landing bias, both $R_0$ (Fig 6D–6F) and final disease incidence (Fig 6G–6I) decrease.

Similar plots are shown for PT in Fig 7, in the non-conditional case in which vector preference does not depend on vector infection status, i.e. when again $v = v_- = v_+$, $\epsilon = \epsilon_- = \epsilon_+$ and $\omega = \omega_- = \omega_+$. As for NPT, $R_0$ increases and final disease incidence decreases with increased landing bias, $v$ (Fig 7A). However, unlike NPT, for PT $R_0$ increases with settling bias, $\epsilon$, and the drop in final disease incidence is less pronounced (Fig 7B). The model exhibits bistability in one-way sensitivity analyses to both landing bias and settling bias. However, and in contrast to NPT, for which the model was bistable for all values of $R_0<1$, there is instead a saddle node

bifurcation at the value $0 < R_0^* < 1$ at which the stable and the unstable (not shown on the graphs) endemic equilibria collide. This means there is only the single disease-free equilibrium whenever $R_0 < R_0^*$. For PT, both $R_0$ and final incidence are irresponsive to the feeding probability, $\omega$, since feeding is required for acquisition/inoculation, and any additional probability of feeding is offset by fewer plants visited per unit time (note how the factors of $\omega_-$ and $\omega_+$ both cancel for PT in Eq (27)). The patterns for Fig 7A and 7B are recapitulated in the two-way sensitivity analyses shown in Fig 7D and 7E.

## Conditional vector preference

Sensitivity scans showing the effect(s) of conditional vector preference–which can only be expressed by PT viruses–are in Fig 8. There is a rapid rise in $R_0$ as both the biases that can potentially be shown by non-viruliferous vectors for infected plants increase; however, there is no change as the corresponding biases shown by viruliferous vectors increase. This is as would be expected from the derivation of $R_0$. The final disease incidence increases but soon levels off with increases in the biases of non-viruliferous vectors for infected plants. With an increasing preference of viruliferous vectors for infected plants, the final disease incidence decreases from a saturation level of 1.0 when this preference is close to 0, i.e. the bias of viruliferous vectors is for healthy plants is increasingly important to reach high disease incidence. Again, two-way sensitivity scans (Fig 8C–8I) reinforce patterns from the one-way analyses, while allowing the effects of interactions between pairs of parameters to be seen. If non-viruliferous vectors are attracted and land or settle upon infected plants ($\nu_-$ and $\epsilon_-$ are both high), but viruliferous vectors are repulsed from infected plants ($\nu_+$ and $\epsilon_+$ both low), this is, unsurprisingly, optimal from the point of view of the virus.

## Interactions between vector population dynamics and disease

Our model allows the maximum vector reproduction rate to depend on the disease status of plants upon which they predominantly feed, via an assumed relative rate of reproduction on infected plants ($\beta$). It also allows us to assume that that more vector flights between plants leads to an increased loss rate of vectors, $\delta > 0$. The paired effects of varying these two parameters simultaneously on final disease incidence is shown in Fig 9A (NPT) and 9B (PT). The overall pattern is similar for both transmission modes. Bistability between the disease-free equilibrium with the vector fixed at carrying capacity and another equilibrium in which the disease has invaded, is again possible. However, for larger values of $\delta$ vectors would die out in the absence of disease (light blue), although sufficiently large values of $\beta$ can then "rescue" both vector and virus populations, allowing disease to persist.

The response of the final disease incidence to the increase in the vector loss rate with the number of plants visited between each successive feed, $\delta$, is shown for fixed values of $\beta$: 1.5 for NPT (Fig 9C) and 2.5 for PT (Fig 9D). These values of $\beta$ were chosen to allow the full range of behaviour to be shown for each mode of transmission and correspond to the horizontal "slices" through Fig 9A and 9B marked by grey lines. The responses of the final disease incidence to the underlying birth rate ($\sigma$) of vectors for different values of the relative birth rate on infected plants ($\beta$) is shown for NPT (Fig 9E) and PT (Fig 9F). In both cases, final disease increases with increasing relative birth rates on infected plants. However, the final disease incidence becomes unstable when a low underlying birth rate is combined with high relative birth rate on infected plants. Bistability is possible for values of $\beta > 1$, where sufficiently large initial disease levels can allow the disease to persist even when the disease-free equilibrium is stable (i.e. when $R_0 < 1$).

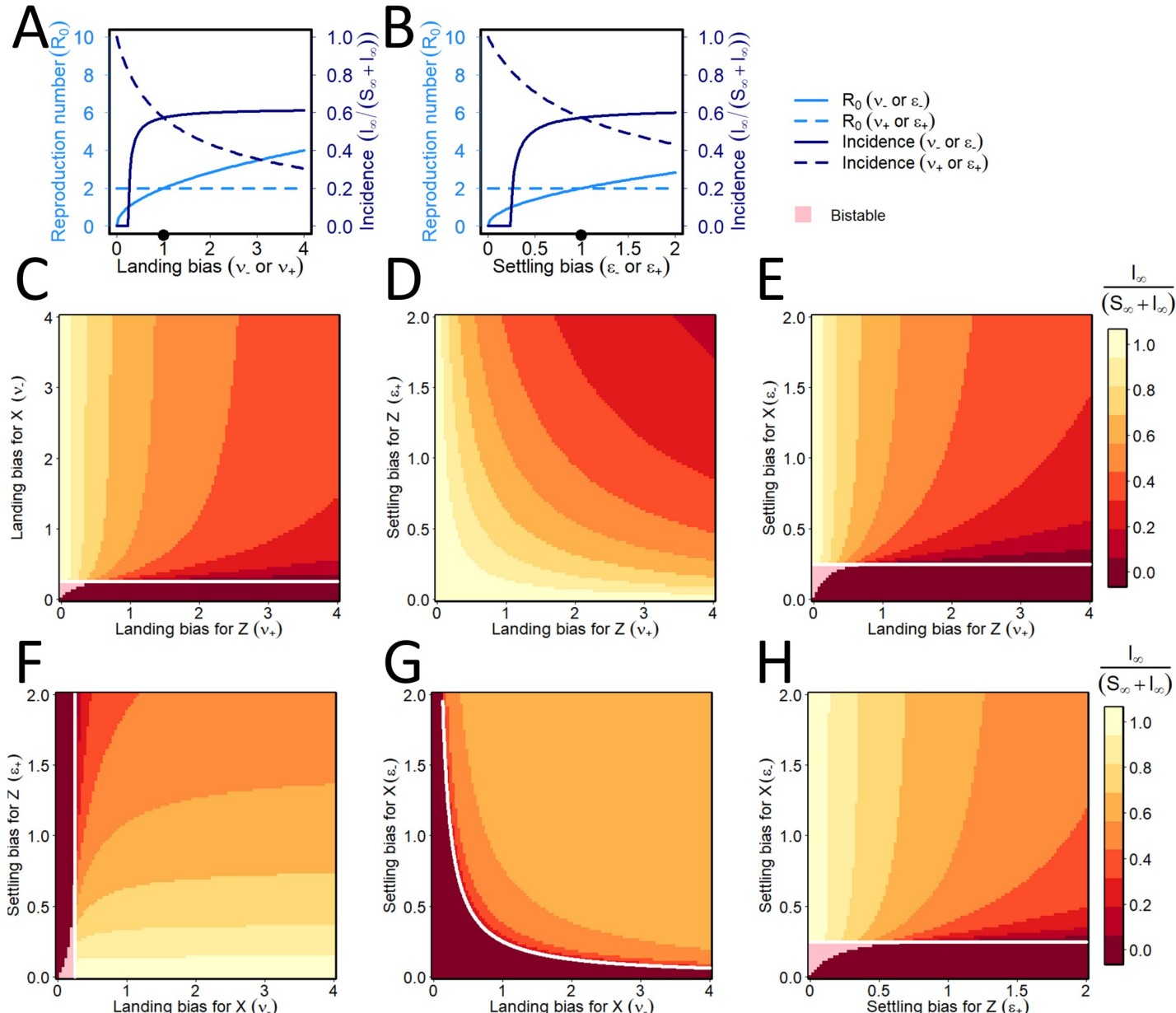

**Fig 8. Persistent transmission: effect of conditional vector preference.** (A)-(B) Responses of the basic reproduction number (light blue) and final disease incidence (dark blue) to changes on the conditional landing (A; $v_-$ and $v_+$) and settling (B; $\epsilon_-$ and $\epsilon_+$) biases of non-viruliferous (solid lines) and viruliferous (dotted lines) vectors. As suggested by Fig 7, both the reproduction number and the disease incidence are irresponsive to changes in both $\omega_-$ and $\omega_+$ for PT, and so these sensitivity analyses are not shown. (C)-(H) Responses of the final disease incidence as pairs of conditional vector preference parameters are simultaneously altered. Again, pairs of parameters for which $R_0 = 1$ are marked with a white curve, with bistability shown by pink shading.

## Interactions between conditional vector preference and population dynamics

For PT viruses, conditional vector preference and population dynamics can interact (Fig 10). For $v_- = 1$, the final disease incidence increases as $\beta$ increases (with diminishing effect for $\beta>1$); as $v_-$ increases above 1 there is little change in the final disease incidence for $\beta>1$ with some decreases for $\beta<1$ (Fig 10A). For $v_+ = 1$, the final disease incidence increases as $\beta$ increases (again with diminishing effect for $\beta>1$); as $v_+$ increases above 1 there is a reduction

 

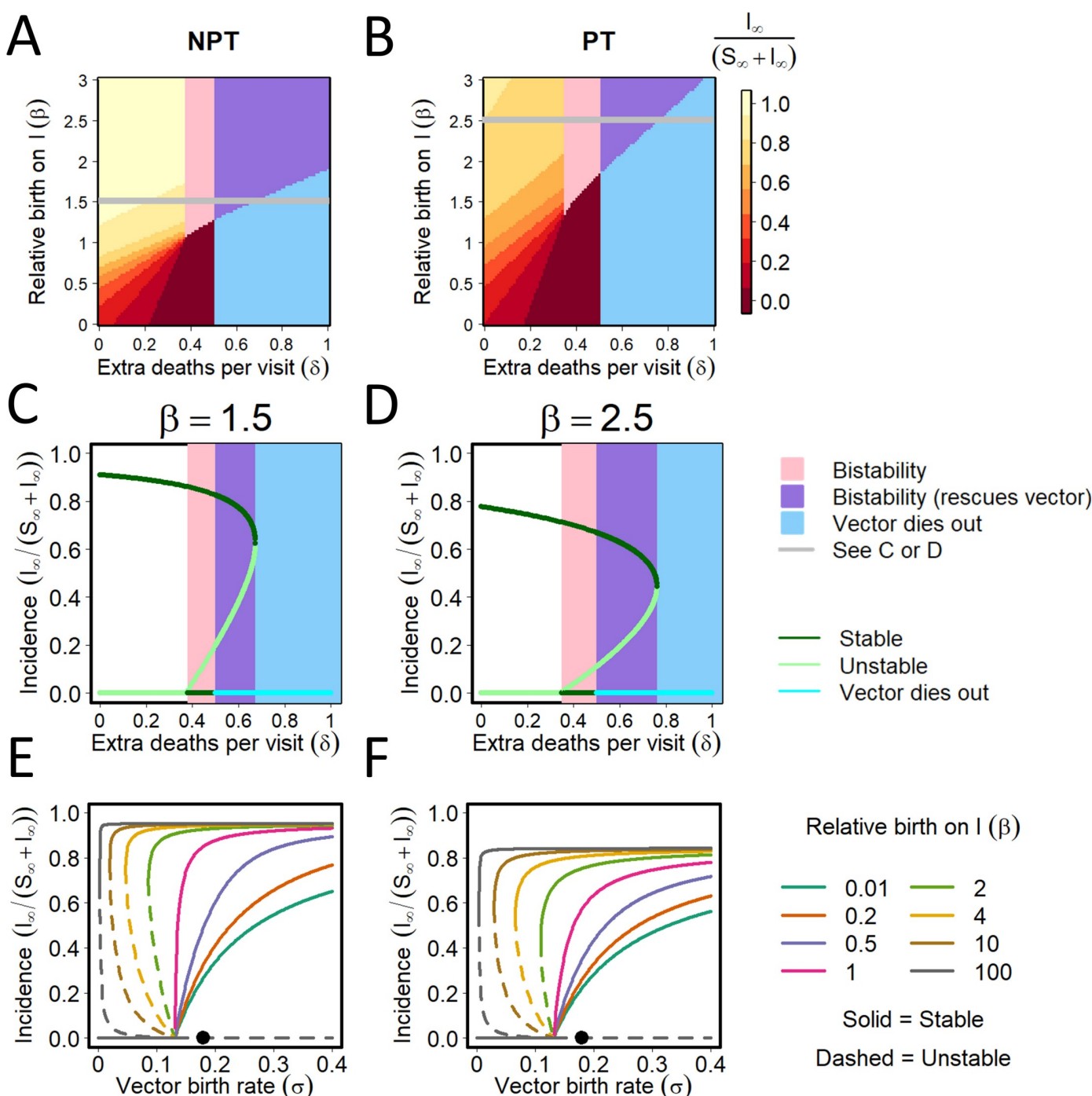

**Fig 9. Interactions between vector population dynamics and disease. (A)-(B):** Final disease incidence in relation to the relative birth rate of vectors on infected plants ($\beta$) and the increase in the vector loss rate with the number of plants visited between each successive feed ($\delta$) for NTP (A) and PT (B). The region of bistability between the disease-free equilibrium with the vector fixed at its carrying capacity and an endemic equilibrium in which the disease has invaded is shaded pink. In the light blue region, for larger values of $\delta$ the vector would die out in the absence of disease, although sufficiently large values of $\beta$ can "rescue" both vector and virus populations (these regions are shown in purple; it is important to recognise that here the vector population is non-zero at equilibrium due to the effect of the virus "rescuing" the vector by promoting enhanced reproduction on infected plants). **(C)-(D):** Final disease incidence in relation to the increase in the vector death rate with the number of plants visited between each successive feed, $\delta$, for a fixed value of the relative birth rate of vectors on infected plants, $\beta$. Responses are shown for different values of $\beta$ for NPT ($\beta = 1.5$) vs. PT ($\beta = 2.5$) to show the full range of behaviour; note these values of $\beta$ correspond to the grey horizontal lines in (A) and (B). **(E)-(F):** Final disease incidence in relation to the underlying birth rate of vectors for NPT (E) and PT (F), for different values of the relative birth rate of vectors on infected plants, $\beta$.

 

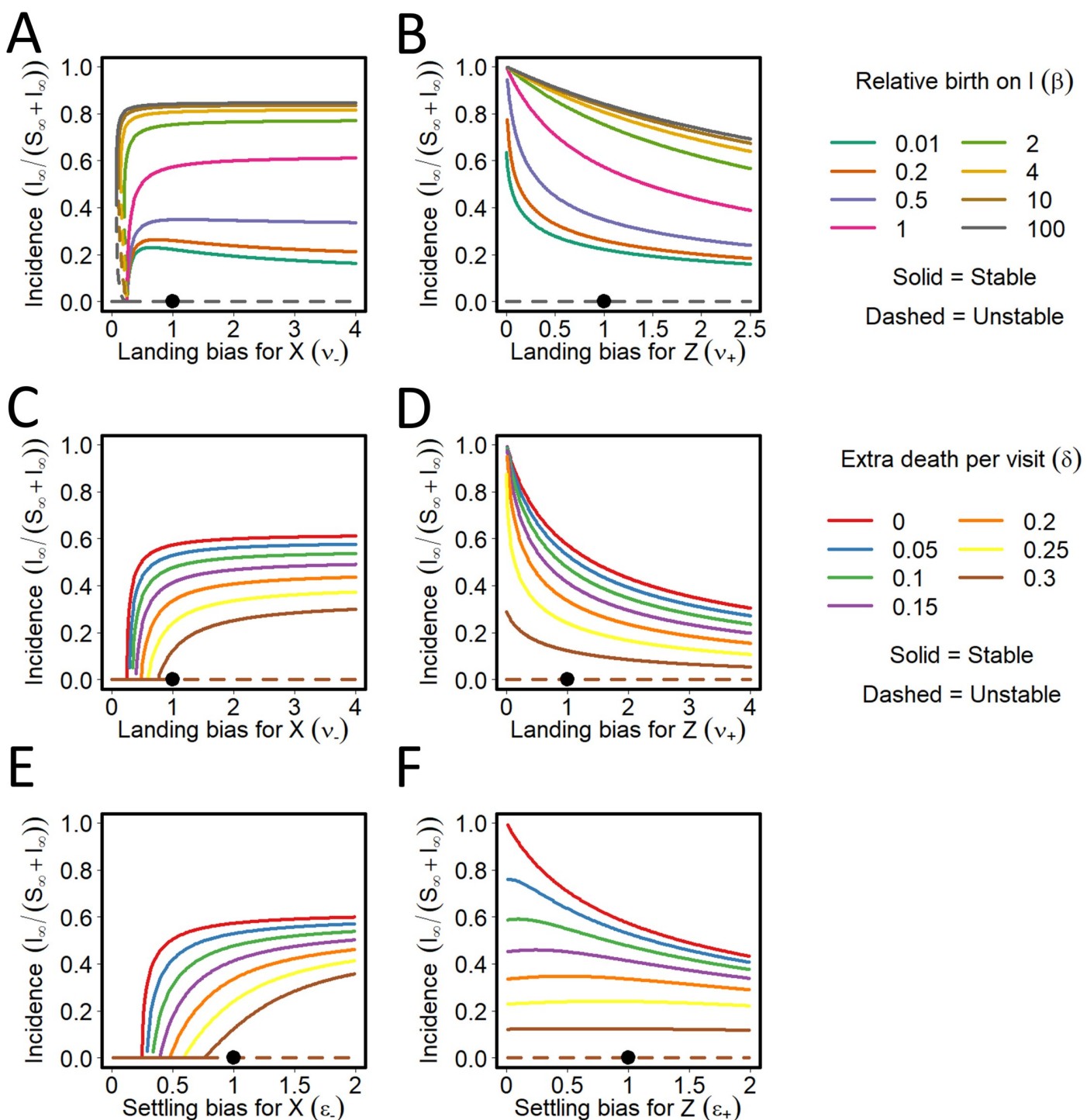

**Fig 10. Persistent transmission: interactions between conditional vector preference and vector population dynamics. (A)-(B)** Final disease incidence in relation to the landing bias for non-viruliferous vectors ($\nu_+$; A) and viruliferous vectors ($\nu_+$; B) for different relative vector birth rates on infected plants ($\beta$). Solid lines indicate a stable equilibrium incidence; dotted lines indicate values for which the equilibrium is unstable; the solid black dot marks the default parameterisation in which there is no bias. **(C)-(F)** Final disease incidence of the landing biases for non-viruliferous vectors ($\nu_+$; C) and viruliferous vectors ($\nu_+$; D) and the setting bias for non-viruliferous vectors ($\epsilon_-$; E) and viruliferous vectors ($\epsilon_+$; F), for different values of the loss rate for vectors of each additional flight between different plants ($\delta$).

in final disease incidence for $\beta < 1$. There is an unstable equilibrium as $v_-$ approaches 0 for $\beta < 1$ (Fig 10B). The effects of the settling bias (i.e. $\epsilon_-$ or $\epsilon_+$) on final disease incidence for different values of $\beta$ are qualitatively similar and are not shown.

The effect on the final disease incidence of the landing biases is shown for different values of the loss rate of vectors for each additional flight between different plants ($\delta$) for non-viruliferous vectors ($v_-$; Fig 10C) and viruliferous vectors ($v_+$; Fig 10D), as well as the setting bias for non-viruliferous vectors ($\epsilon_-$; Fig 10E) and viruliferous vectors ($\epsilon_+$; Fig 10F). Landing and settling biases for non-viruliferous vectors (Fig 10C and 10E) give quantitatively similar final disease incidences for different increased loss rates of vectors ($\delta$), but differ for viruliferous vectors (Fig 10D and 10F). As $v_+$ increases there is a decreasing trend in final disease incidence for all values of δ; however, as $\epsilon_+$ increases there is no trend above $\delta \approx 0.2$ with an indication of a nonmonotonic trend for lower values of $\delta$.

## Interactions between vector preference and population dynamics

The effect on the final disease incidence of the settling bias ($\epsilon_- = \epsilon_+$; A) and landing bias ($v_- = v_+$; B) in relation to relative vector birth rates on infected plants ($\beta$) and for NPT is shown in Fig 11. As there can be no conditional vector preference for NPT, the bias parameters do not depend on the infection status of the vector. At low levels of landing bias close to zero, the final disease incidence is unstable but stable equilibria then decline from a maximum value for $v < 1$ as $v$ increases in a similar way to that seen for conditional preference in PT ($v_+$, Fig 10B). The effect of β in reducing the final disease incidence as landing bias increases is most apparent for $\beta < 1$. The effect of the settling bias ($\epsilon_- = \epsilon_+$, Fig 11A) shows a continuous decrease in final disease incidence as settling bias increases. Similar plots are shown for settling bias (Fig 11C) and landing bias (Fig 11D) in relation to different values of the additional rate of death for vectors of each additional flight between different plants ($\delta$). At sufficiently low values of the settling bias $\epsilon_- = \epsilon_+$ and with $\delta > 0$ the final disease incidence loses stability via a Hopf bifurcation, leading to a sustained periodic solution oscillating around the steady state (Fig 11E and 11F).

## Discussion

### Epidemiological analysis

The model introduced is quite complex, involving parameters describing epidemiological processes, vector life history, and behaviour. However, the complexity of the model is justified in terms of integrating these diverse aspects of the plant-virus-vector interaction. The approach taken in analysing the effects of vector preference on disease epidemiology is novel for plant viruses, although Roosien et al. [47] followed a somewhat similar approach using a simpler mathematical model. Our analysis stresses the importance of the basic reproduction number $R_0$ and the equilibrium or final disease incidence, and how these quantities depend on values of key parameters. It then examines the stability of these equilibria and whether alternative equilibria can be attained depending on initial conditions. Taking an analytical supplemented by a numerical approach enables qualitative as well as quantitative insight. It also enables a clear comparison of differences between NPT and PT. We also provide a user-friendly interface (https://plantdiseasevectorpreference.herokuapp.com/), since in the past we have found this can be very helpful in aiding engagement by non-modellers with modelling results [62].

### Life history and epidemiological parameters

In comparing the basic reproduction number, $R_0$, for NPT and PT, when there is no vector preference, there is a clear divergence for vector death rate ($\alpha$) which gives a higher $R_0$ at low

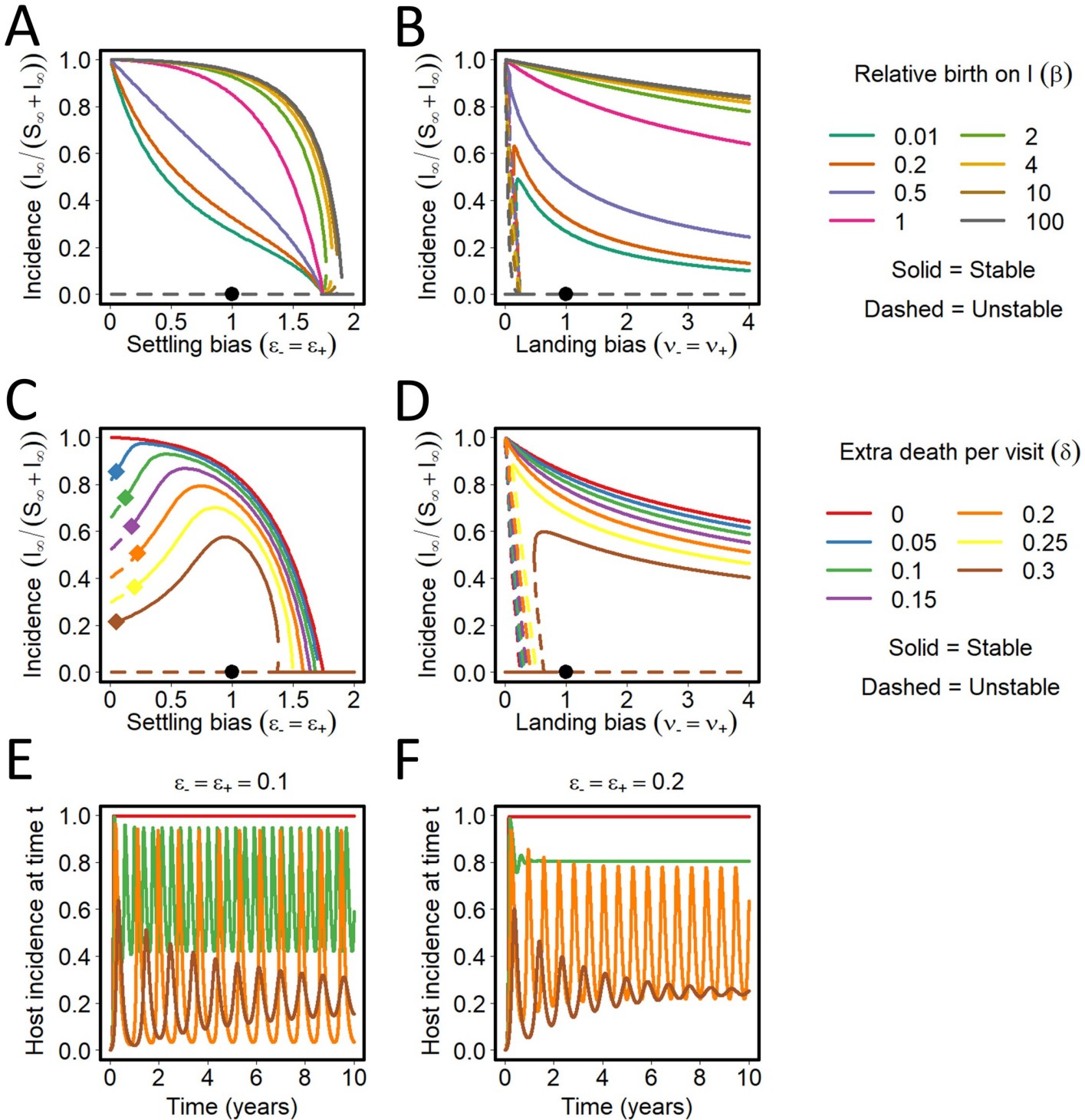

**Fig 11. Non-persistent transmission: interactions between vector preference and vector population dynamics (A)-(B)** Effect on the equilibrium incidence of the settling bias ($\epsilon_- = \epsilon_+$; A) and landing bias ($\nu_- = \nu_+$; B) for different relative vector birth rates on infected plants ($\beta$) for NPT. Solid lines again indicate a stable equilibrium incidence; dotted lines indicate values for which the equilibrium is unstable; the solid black dot marks the default parameterisation in which there is no bias. **(C)-(D)** Effect on the equilibrium incidence of the settling bias ($\epsilon_- = \epsilon_+$; C) and landing bias ($\nu_- = \nu_+$; D) for different values of the additional rate of death for vectors of each additional flight between different plants ($\delta$). At sufficiently low values of the settling bias $\epsilon_- = \epsilon_+$ the disease-present equilibrium loses stability via a Hopf bifurcation, leading to a periodic solution oscillating around the steady state. The bifurcation point is marked with a diamond ($\epsilon_- = \epsilon_+ = 0.058$ when $\delta = 0.05$; $\epsilon_- = \epsilon_+ = 0.134$ when $\delta = 0.10$; $\epsilon_- = \epsilon_+ = 0.197$ when $\delta = 0.15$; $\epsilon_- = \epsilon_+ = 0.236$ when $\delta = 0.20$; $\epsilon_- = \epsilon_+ = 0.222$ when $\delta = 0.25$ and $\epsilon_- = \epsilon_+ = 0.062$ when $\delta = 0.30$). **(E)-(F)** Incidence over time for $\epsilon_- = \epsilon_+ = 0.1$ (E) and $\epsilon_- = \epsilon_+ = 0.2$ (F), showing responses for $\delta = 0$, 0.1, 0.2 and 0.3. In cases corresponding to dotted responses in (C) and (D) sustained periodic oscillations are obtained.

death rates for PT, other parameters held at their default values (Fig 4A vs. Fig 4D). Hence, the need for prolonged settling and feeding, typical of PT, leads to a higher $R_0$ when vector longevity is long. Vector mortality was not explicitly included earlier models [47,48]. For final disease incidence (infected plants as a proportion of total plants at equilibrium) the major divergence between the two modes of transmission arises with the host roguing rate ($\mu$), which can equally be interpreted as additional mortality induced by disease (Fig 4C vs. Fig 4F). A higher final incidence is seen for NPT when $\mu$ is low, other parameters held at their default values. Hence minimal roguing of infected plants in a system affected by an NPT virus would not have a major effect in reducing disease incidence. Another key feature of the parameter $\mu$ is that changing it can induce bistability (Fig 5), potentially making the effects of roguing more unpredictable under certain circumstances. Host mortality–either natural, or additional with roguing or disease-induced–was not included in earlier models targeting vector preference [47,48], despite its effects on disease epidemiology having been shown to be important in past models of virus vector systems without vector preference [53,54], as well as in plant disease models more generally [63].

We do note, however, that we made the very common assumption in plant disease modelling of not attempting to distinguish the regular cycles of planting-harvesting that characterise crop cultivation [64]. At the probable cost of a resulting lack of mathematical flexibility, our model could potentially be included as part of a semi-discrete model which attempted this [65,66]. This would therefore be an interesting extension of our work, albeit one which almost certainly would require extensive numerical simulation. Similarly, were a model with a more careful treatment of individual seasons to be adopted, we could consider the impact of roguing not being followed by replanting. For example, for many annual crops including wheat, potatoes and rice, it would be very unusual for any hosts that were removed by roguing to be replanted within the same season.

A further simplification of our modelling work was to omit the latent period in host plants, mainly since this would require an additional compartment (normally denoted "E" for "Exposed" in the wider mathematical epidemiology literature, although some plant disease modellers prefer "L" for "Latent"). However, this is very common simplification in this type of model [64], and in general has quantitative effects that are relatively well understood (e.g. a reduction in the exponential rate of disease spread early in any epidemic, as well as a small reduction in $R_0$ corresponding to an additional factor due to a small proportion of plants dying before becoming infectious [67]). Nevertheless, since the density of latently infected plants would also affect the choices made by vectors, adding this nuance could be an interesting area for our future work. The same is true of adding a latent period in the vector population. For non-persistent viruses this is clearly not required, since vectors are able to transmit immediately after probing an infected plant, but could be relevant were we to attempt a more careful treatment of different classes of virus we have currently grouped into the "persistent" category (e.g. semi-persistent vs. persistent non-propagative vs. persistent propagative). Following Madden et al. [55] and including a compartment corresponding to infected but as yet not infectious vectors would therefore also be an interesting addition to our work, at least in this case.

We also note that although our treatment accounts for emigration of vectors (via our parameter $\delta$), we make the common modelling simplification of omitting immigration, in part to allow thresholds for disease invasion to be established (since as soon as viruliferous migrant vectors are accounted for, then disease is always able to invade [54,55]). However, transmission via vectors from surrounding fields, volunteer plants, or even weed populations, as well as other related issues such as seasonal immigration and emigration of vectors, can be important in virus disease epidemiology. This is most particularly the case for non-persistently

transmitted viruses, although past modelling work has also targeted other transmission classes [68]. Including these complexities would be an interesting extension for our future work, particularly by linking our treatment of vector preference with other recent work addressing competition between populations of vectors which colonise a particular host population vs. those that only pause briefly on the host population as part of larger-scale migratory behaviour [69]. This would also allow us to begin to address the effects of cooperation–or otherwise–in managing vector densities for epidemics spreading within a community of growers [70]. Since the outcome for any one grower will depend, at least in part, on the controls adopted by other growers, accounting for grower decision making via a game theoretic framework [71] could be a particularly interesting extension of the work presented here.

### Importance of vector preference parameters

A key feature of our model is that it distinguishes vector preferences for the bias to land ($v$) from that to settle ($\epsilon$) on infected plants, as well from that for the probability that the vector feeds ($\omega$). Our basic analysis focused on the case in which the preference does not depend on whether the vector is viruliferous or non-viruliferous (in fact this is the only possible case for NPT viruses). For NPT, the one-way sensitivity analysis showing responses to the bias to land ($v$) recapitulates an already well-known result, i.e. vectors preferring infected plants aids initial invasion of disease by increasing $R_0$, but reduces final disease incidence (Fig 6A). Essentially this is because visits by vectors concentrate on plants that are already infected at high prevalence [44]. However, by separating out the different components of vector preference, our model shows that, as the settling bias and feeding probability for NPT increase for any given landing bias, both $R_0$ and the final disease incidence decrease. This is because vectors which have settled for an extended feed are not able to transmit. An NPT virus able to induce what has been dubbed an "attract-and-deter" plant phenotype in which vectors are attracted to infected plants, but then deterred from settling for an extended feed [18] is therefore expected to be most epidemiologically successful.

For PT, since extended feeds are required to acquire and inoculate the virus, any increase to the settling bias $\epsilon$ leads to the basic reproduction number increasing (Fig 7B), although the final incidence again decreases. The numerical value of the feeding probability $\omega$ is unimportant for PT (Fig 7C), since the formulation of our model properly accounts for the idea that vectors which are engaged in feeding are not able to move between plants and transmit the disease (cf. Eq (27), in which both $\omega$ parameters cancel out of the transmission terms). For both NPT and PT, bistability is possible for landing biases $v<1$ and for a range of settling biases $\epsilon<>1$ (the pink areas on Figs 6A, 6G, 6H, 7A, 7B and 7E). These bistabilities are between a stable disease-free equilibrium and a stable disease-present equilibrium, i.e. occur for $R_0<1$.

The distinction between landing and settling was made in [16], although the model referred only to orientation (rather than settling) bias, and all results presented were based on simulations of an agent-based model. The point has been made [36,37] that interpretation of experimental results depends on whether probing or settling has been reported. In past modelling studies, Roosien et al. [47] did not distinguish between landing and feeding, and Shaw et al. [48] used landing and settling entirely interchangeably. Donnelly et al. [17] did distinguish the two elements of vector preference, but in a model formulation that is heavily conditioned upon the NPT that is that focus of that work. We concur with Fereres and Moreno [33] and Mauck et al. [34,35], that the mode of transmission has different effects on landing, settling, feeding and movement between plants. Our model and results provide–for the first time–a framework to understand the distinct effects of these biases when comparing NPT and PT.

## Conditional vector preference

Conditional vector preference–in which preferences depend on vectors' own infection status–is only relevant for PT. Both $R_0$ and the final disease incidence depend on the landing and settling biases of viruliferous and non-viruliferous vectors, with–again–there being no effect of the probability with which individual vectors feed (Fig 8). There is a rapid rise in $R_0$ as $v_-$ and $\epsilon_-$, the landing and settling biases shown by non-viruliferous vectors for infected plants, increase. There is no change in $R_0$ with the biases shown by viruliferous vectors (i.e. $v_+$ and $\epsilon_+$), since these parameters do not appear in the $R_0$ expression (Fig 8A and 8B). Disease incidence increases but soon levels off with increases in the landing and settling biases of non-viruliferous vectors for infected plants. With increasing landing and settling biases of viruliferous vectors for infected plants, however, the final disease incidence decreases from a saturation level of 1.0 when these biases are close to 0. The bias of viruliferous vectors for healthy plants is increasingly important to reaching a high disease incidence.

Our results are ostensibly rather similar those of Roosien et al. [47] although while those authors did account for conditional vector preference, distinct aspects of vector choice and settling behaviour were confounded into a single "preference" parameter in their model. As noted in [47], there are also similarities to those obtained previously (e.g. [16,44]), but those earlier results were based on uniform preferences for infected plants when these are rare and uniform preferences for healthy plants when infected plants are abundant. Thus, their results are based on the rarity or abundance of infected plants, not on a switching of preferences following virus acquisition as in conditional preference.

## Interactions with vector population dynamics

A key feature of our model is that effect(s) of virus infection on vector fecundity is accounted for explicitly by introducing $\beta$, the proportionate change in the vector birth rate on infected plants (relative to that on healthy plants). Fecundity is a measure of vector performance often used in experimental studies and can increase or decrease on infected plants. Mauck et al. [34] analysed some 224 experimental studies from 55 published papers on vector attraction, settling and feeding, and performance on virus-infected and healthy plants. For vector performance, they found a bias towards positive outcomes for PT viruses and a bias towards negative outcomes for NPT viruses. However, as noted in the Introduction, evidence for this remains mixed across different vector groups, host species, and transmission classes [23,24,26–29,53]. A notable example of how an increase in vector fecundity on virus-infected plants can lead to a marked increase in population size was given by Jiu et al. [72]. The invasive B biotype of the *B. tabaci* complex increased its fecundity on tobacco by 12-fold when infected with tobacco curly shoot virus and by 18-fold when infected with tomato yellow leaf curl virus, whereas the indigenous biotype performed equally on virus-infected and healthy plants. The B biotype displaced the indigenous biotype and was associated with the accelerated spread of the viruses in southern China.

There is no equivalent to the parameter $\beta$ in Roosien et al. [47]. In Shaw et al. [48] vector intrinsic growth rates and carrying capacities were specified separately for healthy and infected hosts, but their results were inconsistent with those of Shaw et al. [73]. Another parameter we introduce is a measure of vector losses from the plant population ($\delta$), arising for example from additional mortality (or other forms of loss) due to the number of flights taken per individual feed. There is no equivalent to this parameter in both of [47] and [48]. We assume that additional mortality applies irrespective of the mode of transmission or whether the vector is viruliferous or non-viruliferous.

The pattern in the parameter space defined by $\beta$ and $\delta$ is similar for both NPT and PT viruses (Fig 9). Bistability between the disease-free equilibrium with the vector fixed at carrying capacity and another equilibrium in which the disease has invaded is once again possible. For larger values of $\delta$ the vector would die in the absence of disease. However, interestingly, sufficiently large values of the relative birth rate on infected plants can then "rescue" both vector and virus populations, allowing disease to persist for certain initial conditions.

The responses of the final disease incidence to the underlying birth rate of vectors for different values of the relative birth rate on infected plants are also quite similar for NPT and PT, with final disease incidence increasing with the relative birth rate on infected plants in both cases. For PT, responses to certain conditional preference parameters were non-monotone for different values of $\beta$ and $\delta$ (e.g. Fig 10A and 10F), indicating that effects on population dynamics of preferential reproduction on infected plants, or of vector losses due to additional mortality from flights, can have relatively complex effects. A similar analysis for NPT–in which vector preference parameters were systematically altered, but in which as before numerical values of "plus" and "minus" parameters were tied together for viruliferous and non-viruliferous vectors–led to rich dynamics. For certain combinations of parameters, we noted the model exhibited sustained oscillations (Fig 11E and 11F). However, we did not analyse the dynamical complexity of the model in any further detail, noting that these periodic solutions required extreme values of underlying parameters.

## Ecological context

The host range of a plant virus reflects considerable ecological complexity [74], as do many other aspects of the host-virus-vector association. This stresses the need to resolve and integrate where possible epidemiology with ecological insights [60,75]. It is clearly difficult to isolate the epidemiological consequences of vector preference from the ecological circumstances found in the field and to test hypotheses on epidemics. Examples of this type of broader ecological context include vector dispersal, the presence of multiple vectors and/or viruses, tritrophic interactions, and direct pest damage caused by vectors. In each case, and as highlighted below, innovative field-based research and/or further modelling will be required to determine the epidemiological significance of the different forms of vector preference.

Dispersal is a key element of insect ecology especially at the local scale [76] where transmission events occur and the movement of insect vectors could be considered in this context. Dispersal strategies are likely to be particularly important in fragmented plant populations [77] where the role of vector preferences in determining non-random movement of vectors in response to host infections status may be critical in disease spread. Plant characteristics are also affected by environmental conditions such as nutrient resources, which can affect vector dispersal and spread of virus [56]. However, whether these characteristics interact with host phenotypes of healthy or infected has not been studied. Dispersal is not modelled explicitly in the present study, but rather implicitly through the number of flights made by a vector for each individual feed. We assume that consequentially there will a loss of vectors from the system, whether due to random movement of non-colonising vectors, additional predation, or additional mortality. The movement of non-colonising vectors out the system is likely to occur as the number of flights increases. Additional mortality may occur due to predation or parasitism, although it may be the well-known alarm signal of vectors under attack that has initiated movement in the first place [56]. The energetics of pollinating insects in relation to the food resource is well-studied [78]. However, the energetic costs associated with number of flights per individual feed with respect to virus transmission and vector preference certainly needs further research.

The model developed is essentially for one virus, one host, and one vector. Co-infection with several viruses is often observed and the interactions between them may affect both the volatile and visual signals picked up by vectors [79,80]. Co-infection is acknowledged to affect underpinning epidemiological dynamics, sometimes in subtle ways [81–83]. Accounting for different plant hosts might also be important. A further model updating the model in [48] has been developed [84], in which conditional vector preference was analysed in multi-host plant communities. Results indicated that non-random associations of viruliferous vectors between preferred and non-preferred host species would occur. This model showed that host diversity led to reduced virus spread, demonstrating the interplay between host diversity and vector behaviour. The competitive effects that may occur with multiple vector species have been addressed, at least from a theoretical standpoint [67,83]. Further research, particularly experimental research, is required on competitive and other interactive effects in relation to vector preference and virus transmission.

Interactions between vectors and non-vector herbivores and pollinators, and other tri-trophic interactions can also affect virus spread [56,85–92]. Links with vector-natural enemy associations have also been confirmed [93,94]. Tomato infected with tomato yellow leaf curl virus changed the host preference of the parasitoid *Encarsia formosa* between Q- and B-bio-types of *B. tabaci*: on infected but not healthy plants the Q-biotype was more attractive to the parasitoid than the B-biotype, due to quantitative differences in volatile profiles. A key question in future research is does a link between vector preference, transmission type, and natural enemies lead to increased virus fitness?

Insect vectors of plant viruses may also be herbivore pests. As such, any preference for healthy or diseased plants may affect the extent of damage as direct pests. A model was developed in which herbivory was linked with a simple epidemiological model [95]. Analysis showed that the virus was better able to persist when the herbivore fed on healthy rather than infected plants. Further, whether the disease establishes an endemic equilibrium depended on the initial conditions, with disease persistence sometimes possible even when the basic reproduction number was under 1. As a result, infected plants were able to persist in the system due to reduced herbivory on these plants but still allowing transmission. A general model investigated the relationship of the basic reproduction number with a further threshold parameter, the consumption number [96]. The behaviour of the model was complex but yielded results showing that "consumer-resource" co-existence could limit the spread of infectious disease. Further experimental and modelling studies are required to disentangle direct pest effects from those resulting from virus infection [97].

## Evolutionary implications

An outstanding question to be asked is what are the evolutionary determinants of vector preference? For example, adaptive dynamics [98] could be used to determine the values of preference parameters that maximise the basic reproduction number. This approach was followed in modelling the evolutionary response of plant viruses to different mechanisms of host plant resistance and disease control [99,100]; and modelling the evolution of virulence/mutualism where there is both horizontal (vector) and vertical (seed/pollen) transmission of plant viruses [101, 102].

A key issue that has not received much theoretical consideration is how vector preference traits, whether non-conditional or conditional have evolved and the extent to which this is due to the vector, the virus, the host, or the parameters defined by their epidemiological interaction. Gandon [11] proposed a theoretical framework applicable to general vectored host-para-site interactions, as a framework for analysing the evolution of preference traits, but with most

emphasis on vertebrate hosts. Preferences for infected hosts were introduced as ratios in the searching efficiencies of vectors carrying, or not carrying, the parasite on infected and noninfected hosts. Although examples of conditional vector preference in insect-vectored plant viruses were introduced, there was no consideration of the differences in NPT and PT transmission, although some of the obtained results for alternative invasion thresholds may be interpreted in terms of transmission mode. The evolutionary analysis used adaptive dynamics and gave insight into why some pathogens have evolved manipulative strategies involving both the vector and the host while others have not.

For plant diseases, such an analysis may prove particularly useful for a scenario in which there is a "new encounter" between a virus and an established host-vector association. Very few studies appear to have looked at this scenario. Porath-Krauss et al. [103] introduce barley yellow dwarf virus into hosts and vectors that the virus had not encountered for thousands of viral generations (estimated as some 250 days), without significant reduction in disease prevalence. In the case of a truly new encounter, we can assume that, prior to first encounter, there would be no vector bias with landing occurring at random. Following Antia et al. [104], we could then consider invasion of the virus as initially unsuccessful at the first encounter because $R_0 < 1$. Subsequently, after several attempts, the virus could "learn" to manipulate the vector and host through manipulation of the landing biases by promoting virus acquisition by non-viruliferous vectors from infected hosts until $R_0 > 1$. Such an analysis, using a simplified version of the current model, will be made in future work by the authors.

## Supporting information

**S1 Appendix. Comparison with previous models of vector preference.**
(PDF)

**S2 Appendix. Deriving the Basic Reproduction Number via the Next Generation Matrix approach.**
(PDF)

**S3 Appendix. Finding model equilibria.**
(PDF)

**S4 Appendix. Number and stability of equilibria.**
(PDF)

## Acknowledgments

NJC thanks John Carr and Ruairi Donnelly for very helpful discussions during initial formulation of the mathematical model.

## Author Contributions

**Conceptualization:** Nik J. Cunniffe, Michael J. Jeger.

**Formal analysis:** Nik J. Cunniffe, Nick P. Taylor, Frédéric M. Hamelin.

**Investigation:** Nik J. Cunniffe, Frédéric M. Hamelin, Michael J. Jeger.

**Methodology:** Nik J. Cunniffe, Nick P. Taylor, Frédéric M. Hamelin.

**Software:** Nick P. Taylor.

**Supervision:** Nik J. Cunniffe.

**Visualization:** Nik J. Cunniffe, Nick P. Taylor.

**Writing – original draft:** Nik J. Cunniffe, Michael J. Jeger.

**Writing – review & editing:** Nik J. Cunniffe, Nick P. Taylor, Frédéric M. Hamelin, Michael J. Jeger.

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
