## [Decision Letter · Decision Letter 0]

17 Nov 2021

Dear Dr. Cunniffe,

Thank you very much for submitting your manuscript "Epidemiological and ecological consequences of virus manipulation of host and vector in plant virus transmission" for consideration at PLOS Computational Biology. As with all papers reviewed by the journal, your manuscript was reviewed by members of the editorial board and by several independent reviewers. The reviewers appreciated the attention to an important topic. Based on the reviews, we are likely to accept this manuscript for publication, providing that you modify the manuscript according to the review recommendations.

Sincerely,

Claudio José Struchiner, M.D., Sc.D.

Associate Editor

PLOS Computational Biology

Tom Britton

Deputy Editor

PLOS Computational Biology

[LINK]

Reviewer's Responses to Questions

**Comments to the Authors:**

Reviewer #1: When I review papers, I typically print them out and read them with pen in hand ready to make corrections. About halfway through the introduction, I put the pen down and simply enjoyed reading the manuscript. The manuscript is well communicated and polished. As a reviewer, I appreciate that the authors took the time to edit the manuscript to a point that it requires little more.

The focus of the manuscript is an important topic in epidemiology and has ramifications for managing the spread of insect transmitted plant pathogens. This paper is dense, and I will acknowledge in the two hours I allotted to complete my review I have not vetted each equation; doing so would take considerable time. Nonetheless, the overall approach used by the authors is appropriate and on the surface the equations and results seem reasonable and appropriate. I do encourage the authors to check their equations for typo’s. It is easy to miss a subscript or that a parameter was raised to an exponent.

I did note that there appears to be an issue with the numbering of the equations. In the “model analysis and numerical approach” section and in the “invasion threshold section” equation 26 is important. However, in my draft there is no equation 26. Rather the equation numbers skip over 26 (goes from 25 to 27). Likewise, I note that earlier in the manuscript equation 3 appears twice. Accordingly, please check the equation numbering.

I thought the “intuitive interpretation” statement that followed equation 36 was excellent. I think a lot of readers will benefit from this.

Overall, I commend the authors for their efforts, I enjoyed reading this.

Reviewer #2: Review of Cunniffe

In this paper, Cunniffe et al describe a model that accounts for how host symptomology may

impact the attractiveness of the plant to insect vectors, as well as other deviations from random

uniform attraction of the insects to the hosts. They use a series of differential equation to

describe their model. They found some rather cool outcomes in terms of long-term oscillations,

bistability, and answers as far as the importance of vector preference for hosts.

Overall, I really like this paper. It's very elegantly addresses a lot of interesting questions all at

once and handily resolves several problems being faced by plant virus epidemiologists for the

past decade. Their online app is a pretty neat feature should help teachers kind of explaining

portions of plant disease epidemiology. I think that they were fairly opaque and how they

described what they were parameterizing their model to. It was not clear if there was a specific

disease in mind, and if there was what disease it was. I would imagine that the state parameters,

transitions between states, and rate parameters would very widely between many different

diseases. For instance, they don't cover if this model is supposed to focus on persistent or nonpersistent viruses, propagative viruses, or even viruses at all. In some ways this is a strength so

that it gives general answers, but it left me wondering how they chose their parameter values,

and how representative these parameters would be for different scenarios or different crops.

Specific Comments:

• Line numbers and double spacing would have helped a ton for reading and reviewing

• They rely on a constant rate of death and harvesting for the plant host and I'm not sure

that that is very realistic for most scenarios. I'm taking a wild guess and saying that they

did this to “keep the model well behaved” But the authors are fairly clever and may have

another solution around this

• They seem to throw quite a bit of shade at another paper in their introduction, although

it's not clear if the paper that they are criticizing is the Shaw 2017/2018 paper or the

Roosien et al. (2013) paper

• “ Error! Reference source not found.” At bottom of page 15

• What are the authors parameterizing their model to? What disease is this supposed to

mimic? Is it supposed to be specifically like a viral disease? Persistent? Non-persistent? I

understand that they are keeping this vague but they are including quite detailed numbers

in their model, a thousand plants as density for example

Reviewer #3: I enjoyed reading this paper, although within the relatively short space of time available for review, I couldn't claim to have really metabolized all of the information it contains. I think the authors have done a good job of packing a large quantity of work into the format of a single paper. Their aims are ambitious here, and while they achieve what they set out to do, the end result is a tightly-packed read that needs to be taken in several bites.

The technical aspects of the paper use familiar techniques from epidemiology and mathematical biology to develop model for two different types of plant virus transmission dynamics. The mathematical approach utilizes linked differential equations and builds on a very strong lineage of research on plant virus dynamics which has been created over several decades by Prof Jeger and colleagues. This is one of the strengths of the paper. The commonality of approach used here with that existing body of work ought to facilitate comparative epidemiology, and I would have liked to see a bit more of that in the current paper. However, (as noted) there is already a lot of material to ingest and comparative work would perhaps best be saved for a purpose-written review? Still, in revision, if the authors see opportunities to highlight further any simple comparative results with previously published analyses I think I would not be alone in finding that useful.

The current introduction does a decent job of setting the scene for the technical sections of the paper, but I wondered if most of the salient information covered in the introduction could be compressed to a tabular supplement that would cross reference aspects of virus transmission biology that have been modeled with relevant literature references, without commentary on whether these previous studies were successful or not? This would give readers a quick look-up table if they are interested in reading the earlier literature for themselves, but would save considerable space in the current paper that could be allocated to a less densely packed exposition of the model.

Building on the issue of the exposition of the model, I would encourage the authors to try to bring out the logic (by logic I mean the deterministic rules that the modeled biology is following) in the model description. The model structure diagram and other visual aids in the relevant figure are excellent, but I found the description of the model parameters and their derivation somewhat dense. The authors state that the model is complex; and it is in the sense that there are many parameters, but it has only four state variables, so in another sense it is relatively simple. Returning to the earlier point about making the exposition of the biological motivations for the model more explicit, exploiting the relative simplicity of the model in terms of the number of state variables, in order to lead the reader through the within-equation complexity could be a useful approach? One way to do this could be to label the equations in (1) (a) to (d) and then explicitly have subsection headings along the lines of "Derivation of equation 1a; the rate of change in susceptible host plants". I think this type of approach would work well in tandem with the interface for the model which the authors have thoughtfully provided. I agree enthusiastically with them that this is an excellent way to allow readers to understand what the model does and how it reflects the underlying biology.

Overall, the paper does a good job of working through the methodology and then the results. There are a few places where I think some rewording would improve the clarity. For example, the second paragraph in the section "Model equilibria" might benefit from some reworking. As currently written, I found it a bit vague for the first sentence or two. Then, when the issue of how many equilibria the model has comes up, I found the current version left me with the impression that the mathematical cart had somehow ended up in front of the biological horse. It's less interesting (at least to me) that solutions compatible with actual biology are among the possible behaviours of the model, than that the model is able to represent biology as it is known to exist; it's a subtle, but I think important, change in focus.

Labeling of the vertical axes in the figures needs to be standardized. For example in figure 5C and 5F the ordinate is labeled "Host incidence". I assume this is supposed to be "Host disease incidence", but in the legend it is referred to as "Disease severity" and in other figures the same quantity (for example in Figure 9 C - F) is labeled as "Incidence" followed by the expression for the diseased fraction of the host population.

Figure 9 contains another element of the paper that (I think) requires a bit of attention. If I have followed the analysis presented, the model contains a section of parameter space in which it illustrates bistability in the absence of the vector. Solutions with this property apparently exist for both PT and NPT types. Since this is, when taken at face value, a surprising outcome, the authors devote some space to offering explanations. It could be an illustration of my lack of comprehension, but I think the story is still missing something. Is the result to be interpreted as being time-bounded in some way? I ask, because, without the presence of a vector I can't think of a biological mechanism through which a vectored virus could be transmitted from infected plants to healthy ones, and unless that happens, won't the virus be eliminated from the system, eventually when the infected plants are removed (by virus-induced mortality, roguing, or harvest?). In other words, in a system with no vector and a composite removal rate greater than zero, doesn't the infected host represent a dynamical dead-end for the virus?

In the section "Ecological context" I think the first paragraph needs some attention. The writing in that paragraph in particular reads as though it was written rather quickly and there are a couple of places where the grammar is questionable. The overall result is that it's not clear (at least to me) what the point is/was.

The second half of the same section (from the final paragraph on the 38th page of the pdf file down to the start of the section "Evolutionary implications" seems mostly irrelevant to the main points of the paper. It deals with contextual results gathered in other studies, but adds little to the explanation of the key findings of the current work. I would recommend the authors remove this section and use the space saved to give a clearer account of the importance of the results from the new model.

Summing up, the paper is an important and timely contribution to the literature on plant virus dynamics. The focus on mechanisms that are studied by empirical biologists in the field and lab should facilitate the interaction between modelers and experimenters in future and the authors deserve a warm thank you from their peers for providing a template for how such detailed modeling can be done. Indeed, the authors make rather little of the conceptual/methodological contribution that the paper makes.

Reviewer #4: Cunniffe and colleagues have tackled a very important problem: the epidemiological consequences of virus manipulation of the host and vector. Most plant viruses are transmitted by insects (primarily Hemiptera) or other arthropods, and these viruses can generally be classified into two broad categories (nonpersistent and persistent) in terms of their transmission properties, or four categories (nonpersistent, semi-persistent, persistent-nonpropagative, and persistent-propagative). The large epidemiological consequences of these categories were previously studied using theoretical SEIR models for the virus and the vector. But these past studies generally did not consider host and vector manipulation by the virus and many other complexities of the pathosystems. For instance, the infection status of a plant can affect the feeding behavior of the vector, virus-status of the vector can affect its behavior. Cunniffe et al. have greatly extended past theoretical work by developing an elaborate deterministic SI compartmental differential equation model for the system, with components for the plant and vector. Emphasis is on how epidemic results differ for nonpersistent and persistent categories.

For the most part, the authors have clearly described the model terms (variables and parameters [and there are MANY parameters]), and have shown detailed results for the nonpersistent and persistent transmission cases. Besides the extensive simulation results, they derived model equilibria and the important basic reproduction number, R0. The results make sense and advance our knowledge of plant virus epidemiology. The work deserves publication.

This is an EXTREMELY long and detailed manuscript. It would be impossible for me to check all of their mathematical results in the allotted time for review. In fact, it would probably take me a year to confirm all of their derivations, especially those presented in the appendices. However, based on past work, I have confidence in the mathematical work by these authors.

Despite the complexity of the model (at least in terms of number of parameters), the authors have made some major simplifications. This is always done with theoretical modeling, of course. But I think the authors should add some discussion to justify these simplifications, and perhaps speculate some more on how their results might be sensitive to the simplifications. This could inspire future work. (The authors have already discussed other complexities of these pathosystems not covered by their model). Given the length of the current manuscript, and the many important results, expansion of the model would only be appropriate for future research and future papers (in my view).

To start with, the authors are using an SI formulation, ignoring the latent (E) and removed (R) states in the plant and vector populations. This is a very common thing to do, especially because it facilitates the calculation of equilibria (steady states, and so on). With this approach, however, removed plants are replaced immediately by new healthy plants. This may be reasonable for epidemics of trees over many years, but with annual crops (wheat infected by barley yellows), removed/dead plants are never replaced within the epidemic. Surely, this merits some discussion.

The authors are grouping three categories of transmission (semi-persistent, persistent-nonpropagative, persistent-propagative) into one broad category. Viruses in their nonpersistent category vary a lot in terms of acquisition and inoculation rates, latent period in the vector, and so on. Is it really reasonable to make conclusions about results for this broad category? I am sure that varying model parameters could handle the different (sub-)categories, except for the latent period in the vector (since an E category for the vector is not considered).

Many epidemics by nonpersistent plant viruses are driven by immigrating/emigrating vectors. For example, aphids from other fields/weeds fly into a soybean field, feed for a few minutes and transmit soybean mosaic virus, and then fly away to another field. Yet, emigration and immigration are not considered in their model. Surely, this also merits some discussion (at least for future work).

**Have the authors made all data and (if applicable) computational code underlying the findings in their manuscript fully available?**

Reviewer #1: Yes

Reviewer #2: Yes

Reviewer #3: Yes

Reviewer #4: Yes

PLOS authors have the option to publish the peer review history of their article (what does this mean?). If published, this will include your full peer review and any attached files.

Reviewer #1: No

Reviewer #2: No

Reviewer #3: **Yes: **Neil McRoberts

Reviewer #4: No

Figure Files:

Data Requirements:

Reproducibility:

References:

---

## [Decision Letter · Decision Letter 1]

15 Dec 2021

Dear Dr. Cunniffe,

We are pleased to inform you that your manuscript 'Epidemiological and ecological consequences of virus manipulation of host and vector in plant virus transmission' has been provisionally accepted for publication in PLOS Computational Biology.

Best regards,

Claudio José Struchiner, M.D., Sc.D.

Associate Editor

PLOS Computational Biology

Tom Britton

Deputy Editor

PLOS Computational Biology

Reviewer's Responses to Questions

**Comments to the Authors:**

Reviewer #1: I enjoyed the manuscript the first time I read it and my thoughts on the current draft are largely the same. This is a dense manuscript; understanding the relationships described in the manuscript requires more time than can be allotted to a review. Nonetheless, the overall approach take by the authors is appropriate and thorough. While I certainly could make some comments about which things to emphasize or deemphasize, much of that is subjective. Accordingly, at this stage I am largely fine with the manuscript as is.

Reviewer #2: Great job tackling the reviewer comments and suggestions! Really nice manuscript, can't wait to see it out

Reviewer #3: Thank you for your prompt and comprehensive response to the questions raised in the previous round of reviewing. Thanks again for this very interesting contribution to the literature.

Reviewer #4: The authors have successfully addressed all issues. This is excellent work and should be published.

**Have the authors made all data and (if applicable) computational code underlying the findings in their manuscript fully available?**

Reviewer #1: Yes

Reviewer #2: Yes

Reviewer #3: Yes

Reviewer #4: Yes

PLOS authors have the option to publish the peer review history of their article (what does this mean?). If published, this will include your full peer review and any attached files.

Reviewer #1: No

Reviewer #2: No

Reviewer #3: **Yes: **Neil McRoberts

Reviewer #4: No

---

## [Editor Report · Acceptance letter]

23 Dec 2021

PCOMPBIOL-D-21-01643R1 

Epidemiological and ecological consequences of virus manipulation of host and vector in plant virus transmission

Dear Dr Cunniffe,

I am pleased to inform you that your manuscript has been formally accepted for publication in PLOS Computational Biology. Your manuscript is now with our production department and you will be notified of the publication date in due course.

With kind regards,

Olena Szabo
